# Experience-dependent shaping of hippocampal CA1 intracellular activity in novel and familiar environments

**Jeremy D Cohen\*, Mark Bolstad, Albert K Lee\***

Janelia Research Campus, Howard Hughes Medical Institute, Ashburn, United States

**Abstract** The hippocampus is critical for producing stable representations of familiar spaces. How these representations arise is poorly understood, largely because changes to hippocampal inputs have not been measured during spatial learning. Here, using intracellular recording, we monitored inputs and plasticity-inducing complex spikes (CSs) in CA1 neurons while mice explored novel and familiar virtual environments. Inputs driving place field spiking increased in amplitude – often suddenly – during novel environment exploration. However, these increases were not sustained in familiar environments. Rather, the spatial tuning of inputs became increasingly similar across repeated traversals of the environment with experience – both within fields and throughout the whole environment. In novel environments, CSs were not necessary for place field formation. Our findings support a model in which initial inhomogeneities in inputs are amplified to produce robust place field activity, then plasticity refines this representation into one with less strongly modulated, but more stable, inputs for long-term storage.

\*For correspondence: cohenj@ janelia.hhmi.org (JDC); leea@ janelia.hhmi.org (AKL)

**Competing interests:** The authors declare that no competing interests exist.

## Introduction

The hippocampus is a major component of the mammalian brain's machinery for learning and memory (*Andersen et al., 2007*). In humans, hippocampal damage results in an inability to convert the ongoing experience of items and events from daily life into long-term memories (*Scoville and Milner, 1957*). In rodents, the hippocampus has been shown to be critically involved in encoding spatial layouts and locations of importance (*O'Keefe and Conway, 1978*; *Morris et al., 1982*). Neural representations of novel (i.e., previously unencountered) environments form rapidly in the rodent hippocampus (*Hill, 1978*; *Wilson and McNaughton, 1993*; *Frank et al., 2004*; *Leutgeb et al., 2004*) and consist of place cells with location-selective 'place field' spiking (*O'Keefe and Dostrovsky, 1971*). Place field activity stabilizes (i.e., becomes more similar across successive traversals of the environment) with experience (*Wilson and McNaughton, 1993*; *Frank et al., 2004*; *Leutgeb et al., 2004*; *Cacucci et al., 2007*), and the persistence of fields over time provides the neural basis of spatial memory (*O'Keefe and Conway, 1978*; *Thompson and Best, 1990*; *Liu et al., 2012*; *Mankin et al., 2012*; *Ziv et al., 2013*).

What processes underlie the formation of hippocampal memory representations? Synaptic plasticity has been strongly implicated in spatial learning-based behaviors (*Morris, 1989*; *Tsien et al., 1996*) as well as changes in place field activity (*Bostock et al., 1991*; *McHugh et al., 1996*; *Mehta et al., 1997*, *2000*, *2002*; *Ekstrom et al., 2001*; *Lever et al., 2002*; *Cacucci et al., 2007*; *Bittner et al., 2015*). However, the role of plasticity in creating new place cell representations is unresolved.

First, is plasticity necessary for place cell activity in novel environments? Pharmacological and genetic manipulations of NMDA receptors (NMDARs) suggest that synaptic plasticity is not needed

for place fields to appear in novel environments (*McHugh et al., 1996*; *Kentros et al., 1998*). However, intracellular recordings suggest that specific, plasticity-inducing events are required to generate each new place field (*Bittner et al., 2015*). In particular, intracellular experiments have shown that large, calcium-mediated complex-spike (CS) events (*Kandel and Spencer, 1961*; *Wong and Prince, 1978*; *Traub and Llinás, 1979*; *Harvey et al., 2009*; *Epsztein et al., 2011*; *Grienberger et al., 2014*), which trigger synaptic plasticity in vitro (*Takahashi and Magee, 2009*) and are sufficient to artificially create place fields, always co-occur with the spontaneous appearance of new fields in familiar environments (*Bittner et al., 2015*). Since, by definition, all place fields in novel environments are new, does their appearance require CSs?

Second, does plasticity strengthen place fields with experience? Within a behavioral session, hippocampal CA1 place field firing rates increase with repeated traversals through each location (*Mehta et al., 1997*, *2000*), which is consistent with a strengthening of the inputs to fields via Hebbian (*Hebb, 1949*) and spike timing-dependent (*Markram et al., 1997*; *Bi and Poo, 1998*) plasticity mechanisms. Across days, though, the opposite occurs, with the average firing rate of CA1 place cells being lower in familiar compared to novel environments (*Nitz and McNaughton, 2004*; *Karlsson and Frank, 2008*). In each case, what changes are occurring to the underlying inputs?

Third, what underlies the stable representation of familiar spaces? The low and sometimes variable (*Mankin et al., 2012*; *Ziv et al., 2013*; *Rubin et al., 2015*) firing rates of CA1 place fields in familiar environments work against having a stable representation of space. However, when a place cell fires, the reliability of where it fires increases with experience (*Wilson and McNaughton, 1993*; *Leutgeb et al., 2004*; *Cacucci et al., 2007*). What changes in inputs produce this overall stability?

Addressing these questions requires measuring the inputs to individual place cells and CSs while animals form new spatial representations, and after environments have become familiar. Therefore, we performed whole-cell intracellular recordings of hippocampal neurons in mice exploring both novel and familiar environments – a comparison which has previously been limited to extracellular recording. Unlike extracellular recording, intracellular recording allows access to inputs via the subthreshold membrane potential ($V_m$), to CSs, and to intrinsic cellular properties such as the AP threshold that shape how inputs are converted into spiking output. Our experiments employed virtual reality-based methods in which animals moved through visually defined virtual environments by running in place (*Hölscher et al., 2005*; *Harvey et al., 2009*; *Chen et al., 2013*; *Ravassard et al., 2013*), which facilitated switching between different environments during intracellular recording. We assessed experience-dependent changes in subthreshold inputs and other intracellular features within a single exploration session in a given environment. We compared findings from sessions in novel environments to those in environments that had become familiar through multiple exploration sessions across several days. The results provide answers to specific plasticity-related questions and, more generally, help to bridge cellular and systems approaches to hippocampal learning. Overall, the findings support a model in which novel environments induce a rapid amplification of input activity in CA1, then the spatial tuning of inputs becomes less strongly modulated, but more reliable, across the entire environment with experience. The result provides the basis for a stable hippocampal representation of familiar spaces.

## Results

### A model of hippocampal spatial memory formation

A model of the formation of place cell representations that fits previous experimental observations is shown in *Figure 1*. It shows the firing rate and subthreshold $V_m$ of a CA1 place cell in both novel and familiar environments as a function of the animal's location. Here, the spatially tuned inputs driving place field spiking are modeled as being initially small, then, due to plasticity, growing with experience (*McHugh et al., 1996*; *Mehta et al., 1997*, *2000*; *Savelli and Knierim, 2010*; *D'Albis et al., 2015*). The observed decrease in firing rate when the environment has become familiar is hypothesized to be due to a lower baseline $V_m$, resulting in reduced spiking output in spite of a larger input-based subthreshold $V_m$ hill in the place field. The lower baseline could arise from the observed increase in inhibitory interneuron firing rates with familiarization (*Wilson and McNaughton, 1993*; *Frank et al., 2004*; *Nitz and McNaughton, 2004*). The inputs driving initial place field firing would be present from the onset of exploration and be based on the existing synaptic weight

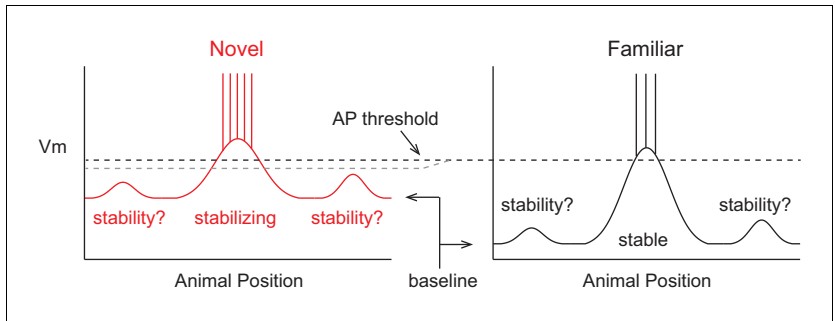

**Figure 1.** A model of intracellular changes underlying the formation of hippocampal CA1 place cell representations. Hypothesized intracellular features underlying place field activity in novel and familiar environments that are consistent with previous studies. Each side represents the membrane potential ($V_m$) as a function of the animal's location in a 1-dimensional environment. With repeated traversals of the environment, inputs underlying place field spiking are strengthened due to Hebbian or spike timing-dependent plasticity, which is reflected in a larger amplitude depolarization of the $V_m$ from the baseline $V_m$ level in familiar environments. The experimentally observed higher AP rate during novel experience is explained by a more depolarized baseline, which could result from the known reduction in CA1 inhibitory interneuron firing rates in novel environments. A lower AP threshold could potentially also contribute to higher AP rates in novel environments (lower, gray dotted line). The spatial tuning of inputs within the place field region is presumed to stabilize with experience, possibly due to plasticity triggered by intracellular complex spikes (CSs). Previous experimental data does not clearly inform what may occur regarding the stability of inputs outside of the field. *Figure 7* shows our results from testing the features of this model with intracellular recordings.

matrix without the need for CSs or plasticity, consistent with the finding that the majority of place fields are present during the first traversal of a novel environment (*Hill, 1978*). Instead, CS occurrence during some of the traversals (laps) across the environment could serve to strengthen inputs for long-term place field stability. This is in agreement with experiments showing that NMDAR-dependent plasticity is not required for place field formation during novel environment exploration, but is necessary for place field stability across days (*Kentros et al., 1998*). The larger subthreshold hill relative to out-of-field inputs would provide the basis for a reliable and long-lasting representation of familiar environments. We tested the features of this model with intracellular recordings of place field activity in behaving animals.

## Subthreshold $V_m$ hill amplitude grows during novel environment exploration

Whole-cell current-clamp recordings of dorsal hippocampal CA1 pyramidal neurons were obtained in adult head-fixed mice as they navigated around novel (NOV) and familiar (FAM) 1-dimensional (1-D) virtual maze environments linked to a spherical treadmill (*Figure 2A–B*; 32 cells from 15 mice; mean ± SD, recording duration: 8.3 ± 7.1 min, recording duration/epoch: 4.3 ± 4.0 min, laps/epoch: 8.9 ± 6.0; maze lengths: 135–240 cm; 'epoch' refers to the entire continuous period in a given maze). Virtual reality (VR) facilitated switching between distinct environments during intracellular recording (*Figure 2C–D*), while the head-fixed behavior in the track-like mazes yielded repeated exposures to all spatial locations within an environment without backtracking. Each cell's location-based activity with respect to a common reference (the fixed-location primary reward zone) was uncorrelated across distinct mazes (*Figure 2—figure supplement 1*), suggestive of global remapping (*O'Keefe and Conway, 1978*; *Muller and Kubie, 1987; Leutgeb et al., 2005*) between the virtual environments. The animals' licking shifted from occurring just after reward delivery at the primary reward zone in novel environments to occurring just before delivery (i.e., predictively) in familiar environments (*Figure 2E* and *Figure 2—figure supplement 2A*), providing behavioral evidence of spatial learning.

We first checked the amplitude of the main subthreshold $V_m$ hill during individual maze epochs for evidence of increases in synaptic strength. In novel environments, the peak amplitude increased between the initial (1–2) and late (6-end) laps (*Figure 3A*, left, 3B), similar to the number of laps

over which substantial changes in place fields have been observed in extracellular studies (*Mehta et al., 2000*; *Ekstrom et al., 2001*). This was the case when considering the peak subthreshold $V_m$ within the field determined from the average subthreshold $V_m$ as a function of location over the entire epoch (median ± SE, NOV initial laps: 1.9 ± 0.5 mV, late laps: 3.1 ± 0.6, p=0.003, n = 14, *Figure 3A*, left), or, to allow for spatial jitter, the peak in each lap irrespective of location ('lap peak', NOV initial laps: 3.0 ± 0.4 mV, late laps: 4.0 ± 0.6, p=0.004). The increase in amplitude is consistent with a strengthening of inputs to place fields via Hebbian, spike timing-dependent, or other (*Golding et al., 2002*; *Dudman et al., 2007*) mechanisms of synaptic plasticity, and indicates rapid learning in CA1 in novel environments. In contrast, no increases in amplitude were observed within familiar mazes (p=0.47, *Figure 3A*, right; lap peak: p=0.70).

To further address possible mechanisms, we examined the subthreshold $V_m$ amplitude with respect to the first lap with spiking in the place field. We decomposed the novel maze increase in peak subthreshold $V_m$ amplitude within the place field region (NOV initial laps: 3.0 ± 0.7 mV, late laps: 4.8 ± 0.8, n = 9, p=0.039, *Figure 3B*; in contrast, FAM initial laps: 3.9 ± 1.3, late laps: 3.0 ± 0.6, n = 10, p=0.63) into two parts. For the subset of novel epochs in which place field spiking did not occur in the first lap, the peak subthreshold $V_m$ in the place field increased sharply from the laps before to the laps after spiking began (*Figure 3C–D and E*, left). However, there was no consistent increase after the first spiking lap, including cases in which place field spiking occurred in the first lap of the epoch (First-Active lap: 4.0 ± 0.9 mV, Post' laps: 4.1 ± 0.5, n = 9, p=1.0, *Figure 3E*, right). The sharp increase suggests mechanisms of plasticity that do not rely on postsynaptic spiking (*Golding et al., 2002*; *Dudman et al., 2007*). The lack of increase in peak subthreshold $V_m$ amplitude after the first spiking lap was also observed for place fields in familiar mazes (p=0.42). In all five cases in which place field spiking was not present in the first lap of a novel epoch, the mean $V_m$ in the laps before spiking was greater inside versus outside the eventual place field (*Figure 3F*; also see *Figure 3C–D*, bottom right). This suggests that inputs were biased towards the locations of the place fields before spiking occurred.

## Subthreshold $V_m$ hill amplitude is smaller in familiar environments

We then compared the amplitude of subthreshold $V_m$ hills in familiar versus novel environments. First, in agreement with previous extracellular recordings from freely moving rats (*Nitz and McNaughton, 2004*; *Karlsson and Frank, 2008*), the peak firing rate of active cells was higher in novel environments (peak of average firing rate as a function of location over the entire epoch, NOV: 5.9 ± 2.1 Hz, FAM: 1.7 ± 0.5, n = 10 NOV-FAM maze pairs with at least one epoch being active, p=0.004, *Figure 3G*; lap peak: NOV: 13.0 ± 2.3 Hz, FAM: 7.7 ± 2.1, p=0.004, *Figure 3—figure supplement 1B*), and the proportion of cells that were active was greater (69% of NOV epochs and 47% of FAM epochs).

According to the model in *Figure 1*, the strength of place field inputs should either continue to increase or remain strengthened with additional experience, with the lower firing rates in familiar environments being explained by a more hyperpolarized baseline $V_m$ and/or higher action potential (AP) threshold. However, we found that the spike threshold (*Figure 4A*) and baseline (*Figure 4B*) did not differ between novel and familiar environments. Instead, the amplitude of the peak subthreshold $V_m$ was smaller in familiar environments (NOV: 2.4 ± 0.4 mV, FAM: 1.8 ± 0.2, n = 14 active and silent NOV-FAM maze pairs, p=0.042, *Figure 3H*; lap peak: p=0.002, *Figure 3—figure supplement 1C*; active maze pairs subset: NOV: 3.2 ± 0.4 mV, FAM: 1.9 ± 0.2, n = 8, p=0.023; note that including only active pairs controls for the lower proportion of active cells in familiar mazes; peak in place field: NOV: 2.5 ± 0.4 mV, FAM: 1.6 ± 0.1, n = 10 maze pairs with at least one epoch having a place field, p=0.004, *Figure 3I*). Furthermore, there was no sign that any part of the increase in amplitude during novel exploration remained by the time environments became familiar (p=0.52, *Figure 3J*, left). The second highest subthreshold $V_m$ peak in each environment was not, however, larger in novel compared to familiar environments (NOV: 1.4 ± 0.1 mV, FAM: 1.4 ± 0.1, n = 14, p=1.0).

Together, these results indicate that hippocampal CA1 neurons responded uniquely to novelty with enhanced inputs, leading to higher firing rates than in familiar environments. Another likely contributor to the increased spiking in novel environments was larger temporal fluctuations of the subthreshold $V_m$, reflecting the instantaneous input activity about the mean hill amplitudes described so far. Single place field traversals in novel and familiar mazes could display robust theta (~5–10 Hz)

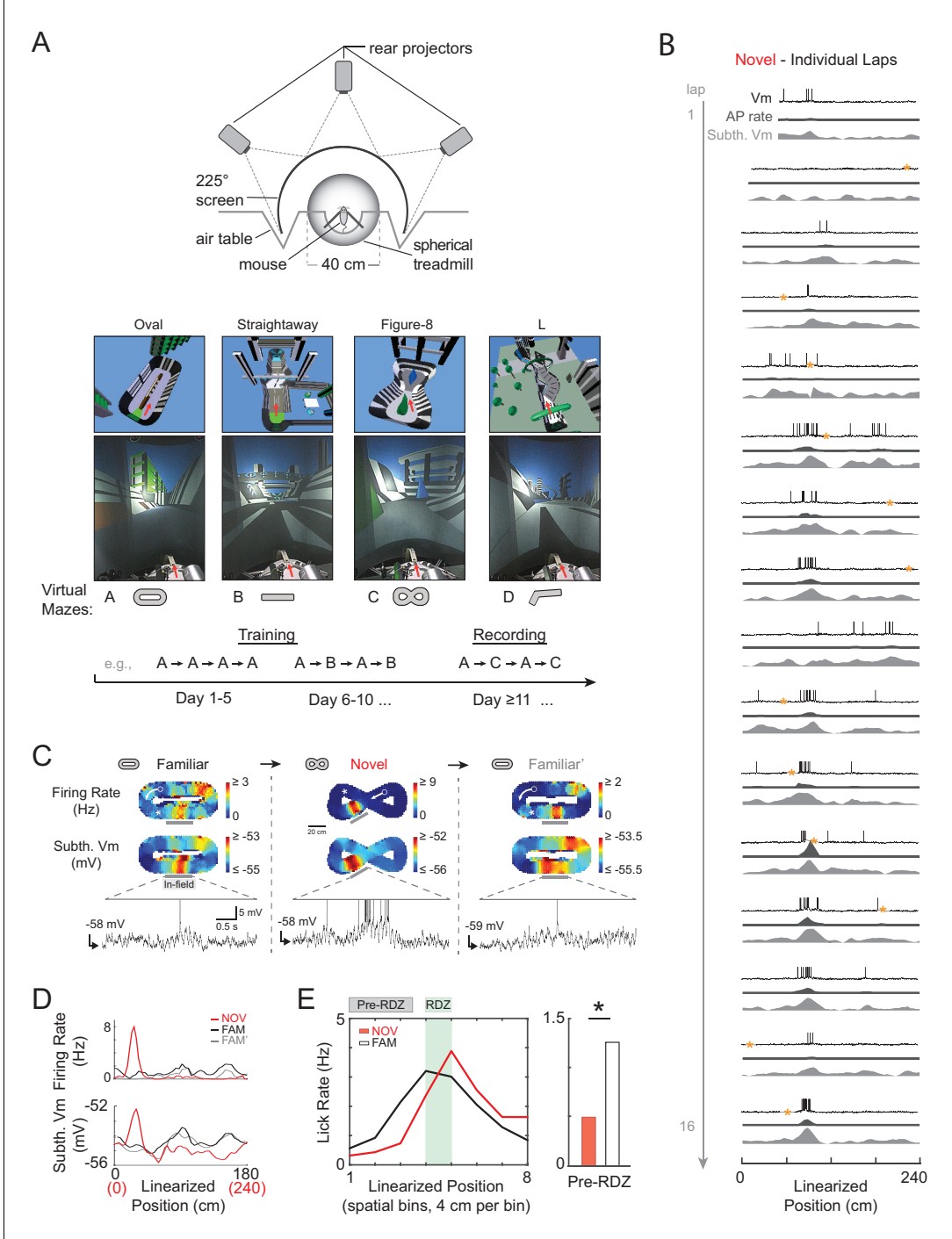

**Figure 2.** Whole-cell intracellular recording of mouse hippocampal CA1 neurons in novel and familiar virtual maze environments. (A) Top: Top view of virtual reality apparatus showing mouse on a spherical treadmill surrounded by an image of a maze environment projected onto a cylindrical screen (225° arc). Middle: The four virtual maze environments used in the study. Overhead view of 3-D scene models of the mazes (above). Photos of the rear-projected virtual mazes on the cylindrical screen taken from above and behind the animal position (below). Arrow shows location and perspective of the animal in the virtual maze (above) and on the spherical treadmill (below). Bottom: Example behavioral training and recording protocol. (B) Example whole-cell intracellular recording in a novel maze. $V_m$ (top), AP rate (middle), and subthreshold $V_m$ (bottom) for the first 16 laps in linearized position coordinates. Yellow asterisks mark times (every 20 s) when current injection was applied to probe series and input resistance and evoked spiking response. Note: 'Subthreshold $V_m$' refers to the $V_m$ after APs and CSs have been removed. (C) Activity of same cell shown in (B) as the animal first explored a familiar maze (left), then a novel maze (middle), then was re-exposed to the initial familiar maze (right). Overall AP rate (top) and subthreshold $V_m$ (middle) in each of these three epochs. Note: 'Epoch' refers to the entire continuous period in a given maze. White arrow: Running

*Figure 2 continued on next page*

*Figure 2 continued*

direction. White asterisk: Primary reward location. Example $V_m$ traces (bottom) from single traversals (gray bars) through place fields in each epoch (APs truncated). (D) Overall epoch activity from (C) in linearized coordinates. (E) Left: Lick rate in spatial bins immediately surrounding primary reward zone (RDZ) in novel and familiar mazes. Right: Predictive licking behavior in spatial bins immediately preceding RDZ (Pre-RDZ lick rate in Hz) in familiar versus novel mazes. *p<0.05.

The following figure supplements are available for figure 2:

**Figure supplement 1.** AP rate and subthreshold $V_m$ activity aligned to the primary reward zone is not correlated between novel and familiar mazes.

**Figure supplement 2.** Increased predictive licking behavior with experience.

fluctuations (*Figure 3—figure supplement 2A*). As in freely moving rats (*Lee et al., 2012*), the theta-band power was slightly positively correlated with mean subthreshold $V_m$ (*Figure 3—figure supplement 2B*), implying that peak theta power should be somewhat higher in novel environments. Indeed, we found slightly increased in-field theta power (5–10 Hz $\text{power}^{1/2}$ in mV, NOV: $0.79 \pm 0.09$, FAM: $0.68 \pm 0.06$, n = 14, p<0.001), as well as gamma power (25–100 Hz $\text{power}^{1/2}$ in mV: NOV: $0.33 \pm 0.03$, FAM: $0.30 \pm 0.02$, p=0.001), and $V_m$ standard deviation (NOV: $1.31 \pm 0.12$ mV, FAM: $1.19 \pm 0.10$, p=0.011) in novel mazes (*Figure 3—figure supplement 2C–E*). Although small in amplitude, the extra instantaneous bump to the $V_m$ hills at the peaks of these fluctuations likely adds to the enhanced spiking responses during novel exploration.

Hippocampal firing rates have been shown to increase with running speed (*McNaughton et al., 1983*). Here, the larger subthreshold responses in novel mazes were not attributable to differences in running behavior. Animals ran at similar overall speeds (NOV: $16.6 \pm 1.3$ cm/s, FAM: $15.8 \pm 0.7$, n = 14, p=0.63, *Figure 3—figure supplement 1D*) and paused (periods >1 s with speed <2 cm/s) with similar frequency (p=0.078, *Figure 3—figure supplement 1E*) in novel and familiar mazes. Furthermore, the instantaneous speed of the animal was not correlated with the instantaneous subthreshold $V_m$ across novel and familiar epochs (median $\pm$ SE of Pearson's r value per epoch: $0.01 \pm 0.02$, p=1.0). However, the differences in reward-related predictive licking (*Figure 2E*) and speed profile (see below) in novel versus familiar mazes provide behavioral evidence that the animals recognized the novelty and familiarity of the environments in VR.

## Somatic excitability does not contribute to enhanced responses in novel environments

We next investigated cellular properties for possible contributions to the enhanced subthreshold and spiking responses in novel mazes. Differences in postsynaptic excitability could affect the magnitude of depolarization elicited by a given presynaptic input, as well as affect the spiking response to a given subthreshold $V_m$ depolarization. However, somatic excitability did not differ between novel and familiar environments according to four measures: spike threshold, baseline (resting) $V_m$, input resistance, and evoked spike count (*Figure 4A–D*).

As already described, the AP threshold did not differ between novel and familiar environments (NOV: $-47.5 \pm 1.4$ mV, FAM: $-48.0 \pm 1.7$, n = 7, p=1.0, *Figure 4A*; also see *Figure 4—figure supplement 1A*). Also, as described, the baseline $V_m$ – which can relate to excitability as well as to inputs – did not differ (NOV: $-55.5 \pm 1.3$ mV, FAM: $-56.6 \pm 1.2$, n = 14, p=0.63, *Figure 4B* and *Figure 4—figure supplement 1B–D*). A closer analysis of the baseline revealed no evidence of a role in the larger subthreshold or spiking responses during novel exploration. First, there were no transient changes in baseline found around the time of the switch between mazes (*Figure 4—figure supplement 1E*). In particular, there was no transient depolarization of the baseline that could trigger a sustained increase in activity during novel exploration. Furthermore, considering each neuron individually, the difference between a cell's baseline in novel and familiar epochs could not account for the difference in peak subthreshold $V_m$ (p=0.37, *Figure 4—figure supplement 1F*). A higher somatic membrane input resistance ($R_N$) in novel mazes could cause larger subthreshold $V_m$ hills in response to otherwise similar inputs. However, somatic $R_N$ did not differ between novel and familiar mazes (p=0.30, *Figure 4C*), and any differences within individual neurons could not account for the larger peak $V_m$ responses in novel environments (p=0.43, *Figure 4—figure supplement 1G*). Lastly,

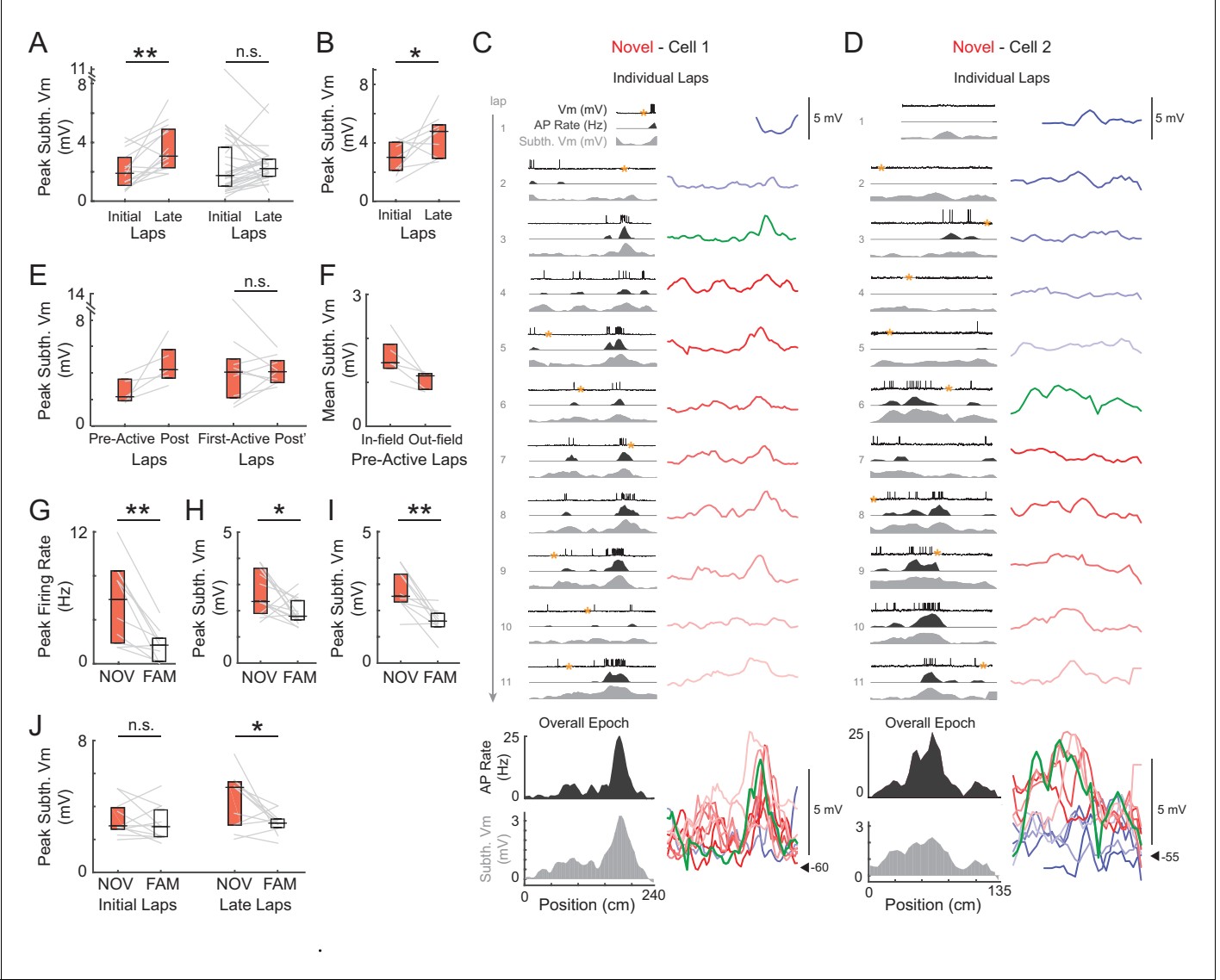

**Figure 3.** Subthreshold $V_m$ depolarizations underlying hippocampal CA1 place fields grow in novel environments and are larger in novel compared to familiar mazes. (A) Peak subthreshold $V_m$ in the initial laps 1–2 versus the late laps 6-end for all novel (NOV, left) and familiar (FAM, right) mazes. The peak refers to the peak value within the field determined from the epoch-averaged subthreshold $V_m$ as a function of location. Here and elsewhere: Box marks 25-75th percentile values, horizontal line marks median. (B) Peak subthreshold $V_m$ within the (spiking) place field in the initial versus late laps for all place fields in novel mazes. (C)-(D) Examples of place fields that formed in a lap later than the first lap in a novel maze. Subthreshold $V_m$ for each lap at expanded (and fixed) voltage scale (right) and overlaid (bottom right). (E) Left: Peak subthreshold $V_m$ within the place field in the laps before (Pre-Active) versus after (Post) the first lap with spiking in the field. This includes only those place fields in novel mazes that formed in a lap later than the first lap. Note that Post includes the first lap with spiking in the field. Right: Peak subthreshold $V_m$ within the place field in the first lap with spiking in the field (First-Active) versus in the laps afterwards (Post'). This includes all place fields in novel mazes, including those that first spiked within the field in lap 1. In (C, right) and (D, right): Pre-Active laps (shades of blue), First-Active lap (green), Post' laps (shades of red), Post laps (green and shades of red). (F) Mean subthreshold $V_m$ inside versus outside the field in the laps before place field spiking for the novel maze place fields in (E, Left). (G) Overall epoch peak firing rate for all recorded pairs of novel and familiar mazes (NOV-FAM maze pairs) in which cell was active in at least one of the mazes in the pair. (H) Overall epoch peak subthreshold $V_m$ for all NOV-FAM maze pairs whether or not cell was active or silent in either maze. (I) Same as (H) except for NOV-FAM maze pairs in which the cell had a place field in at least one of the mazes in the pair. (J) Peak subthreshold $V_m$ within the $V_m$-defined field in the initial laps 1–2 (left) and late laps 6-end (right) of all NOV-FAM maze pairs. P-values for paired comparisons from Wilcoxon signed-rank tests. *p<0.05; **p<0.01.

The following figure supplements are available for figure 3:

**Figure supplement 1.** Higher AP rate and larger amplitude subthreshold $V_m$ responses during novel exploration are not due to differences in running speed or stopping behavior.

*Figure 3 continued on next page*

*Figure 3 continued*

**Figure supplement 2.** Intracellular membrane potential oscillations are larger during novel maze exploration.

subthreshold $V_m$ hills of a given amplitude could potentially result in different spiking output even with a similar AP threshold (e.g., due to an increased propensity to burst). However, the number of spikes evoked by depolarizing current steps (0.1–0.2 nA, 0.1 s) did not differ between novel and familiar environments (p=0.55, *Figure 4D*).

Although we found no changes in somatic excitability between novel and familiar mazes during an individual recording, we asked how intrinsic properties, such as the AP threshold, might regulate

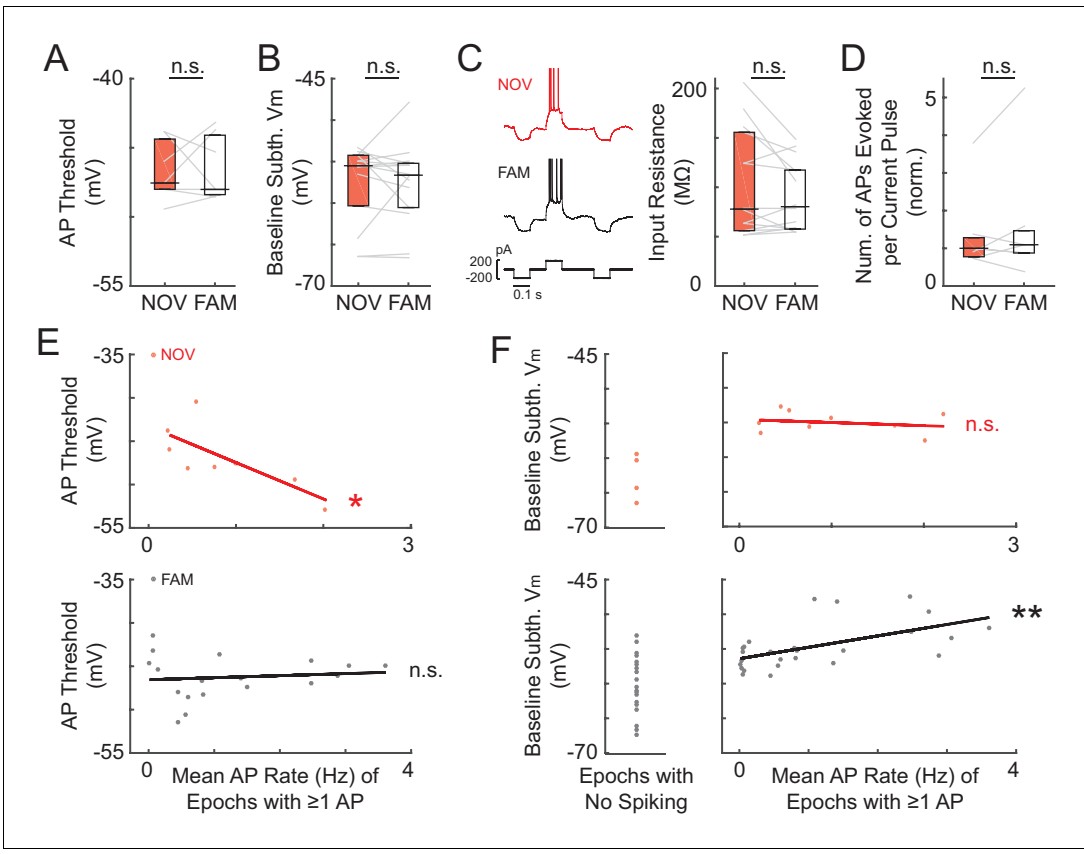

**Figure 4.** Somatic excitability does not differ between novel and familiar mazes. (A) Mean AP threshold for all cells recorded in a NOV-FAM maze pair that were active in both mazes. (B) Overall epoch baseline subthreshold $V_m$ for all cells recorded in a NOV-FAM maze pair. (C) Left: Example intracellular current injection sequence and responses (to probe $R_N$ and evoked firing) from a NOV-FAM maze pair. Right: Mean somatic $R_N$ for all cells recorded in a NOV-FAM maze pair. (D) Mean number of APs evoked by the positive step in the current injection sequence (normalized to median value for each step size). (E) Mean AP threshold (mV) versus AP rate (Hz) for all NOV (top) and FAM (bottom) maze epochs with ≥1 AP. (F) Left: Baseline subthreshold $V_m$ (mV) for all NOV (top) and FAM (bottom) epochs with no spontaneous APs. Right: Baseline subthreshold $V_m$ versus mean AP rate for all NOV (top) and FAM (bottom) epochs with ≥1 AP. The baseline $V_m$ was more depolarized in epochs with ≥1 AP than completely silent epochs (NOV: p=0.002, FAM: p<0.001). *p<0.05; **p<0.01.

The following figure supplement is available for figure 4:

**Figure supplement 1.** No differences in AP thresholds, somatic baseline $V_m$, or input resistance can account for the larger amplitude subthreshold $V_m$ responses in novel environments.

spiking activity across different recordings. In agreement with previous findings (*Epsztein et al., 2011*), we found that the AP threshold was negatively correlated with the mean AP rate in novel mazes (p=0.041, *Figure 4E*, top). Interestingly, no such correlation was found in familiar mazes, for which the mean AP rate was instead positively correlated with the baseline $V_m$ (p=0.003, *Figure 4F*, bottom right). In addition, the baseline $V_m$ was more depolarized in epochs with $\geq 1$ AP than in epochs with no spontaneous activity for both novel and familiar mazes (NOV: p=0.002, FAM: p<0.001, *Figure 4F*). These results suggest that in a novel environment, intrinsic cellular excitability helps to determine the initial place field representation. Then, when the environment has become familiar, inputs can drive the stored spatial representations in CA1 irrespective of excitability.

## Spatial tuning of subthreshold $V_m$ stabilizes with experience

As increases in the strength of place field inputs did not appear to be sustained, we looked for other evidence of lasting spatial learning. Stabilization of spatially tuned firing with experience is a sign of hippocampal learning (*Wilson and McNaughton, 1993*; *Mehta et al., 1997*, *2000*, *2002*; *Kentros et al., 1998*; *Leutgeb et al., 2004*; *Frank et al., 2004*; *Cacucci et al., 2007*). Experience-dependent changes in the spatial tuning of inputs have not previously been studied with intracellular recording. Moreover, using extracellular recording, changes cannot be inferred where there is little or no spiking, such as regions outside of place fields and in silent cells. Therefore, we analyzed the tuning of inputs across the whole environment as reflected in the shape of the entire subthreshold $V_m$ curve versus location (*Figure 1*), and changes in this tuning with experience.

We assessed spatial tuning as a function of experience by computing the linear correlation of AP rate or subthreshold $V_m$ activity in each lap with the average activity over the entire epoch ('overall epoch') (*Figure 5A* and *Figure 5—figure supplement 1A*), where each correlation was computed across all spatial bins (i.e., the entire extent of the maze). To begin with, high or low spatially correlated activity could be observed in novel or familiar mazes (*Figure 5—figure supplement 2*), and an individual cell's correlation score in novel mazes did not predict its score in familiar mazes (or vice versa) for AP rate (p=0.16) or subthreshold $V_m$ (p=0.83). We therefore compared spatial correlation scores for novel versus familiar epochs independent of paired exposures. All epochs were included, with the following exception. In our experiments, for each recording, the animal was generally first exposed to a familiar maze followed by a novel or other familiar maze. In many cases (10/32 cells), this first familiar maze recording was also the first exposure to any maze for the animal on that day. We suspected that spatial correlations could be disrupted in these first familiar epochs due to non-specific behavioral factors (*Figure 2—figure supplement 2B*), thus partially masking differences from novel epochs. More generally, this is a factor that may be useful to consider in future VR studies. With all first epochs of the day (whether familiar or novel) excluded (*Figure 5B*; see *Figure 5—figure supplement 1B* for the same analysis including all epochs; note, excluding the initial epoch of each day did not alter any of the previous findings such as the difference in peak AP rate and subthreshold $V_m$ amplitudes, *Figure 5—figure supplement 3*; similarly, excluding the region surrounding the primary reward location did not alter these results), we observed that AP rate correlation scores were, again, generally high and similar (p=0.29) in familiar (Pearson's r: 0.65 ± 0.14) and novel (0.58 ± 0.09) mazes (*Figure 5B*, top).

In contrast, the subthreshold $V_m$ spatial correlation scores were significantly lower in novel than familiar mazes (NOV: 0.58 ± 0.12, FAM: 0.69 ± 0.03, p=0.003, *Figure 5B*, middle left). Moreover, the $V_m$ correlation scores were lower in the initial laps (p=0.003, *Figure 5B*, middle right), then significantly increased across laps within a novel but not familiar epoch, reaching close to the familiar values in later laps (*Figure 5B*, middle right). This progressive stabilization of spatial tuning with experience is an intracellular signature of the establishment of a long-term memory trace.

## Spatial tuning of subthreshold $V_m$ stabilizes across the entire environment

We then asked where this progressive stabilization was occurring. Extracellular studies have shown that CA1 place field spiking stabilizes with experience (*Wilson and McNaughton, 1993*; *Frank et al., 2004*; *Leutgeb et al., 2004*; *Cacucci et al., 2007*). Is the stabilization of spatial tuning we observed in CA1 occurring for its inputs across the whole environment?

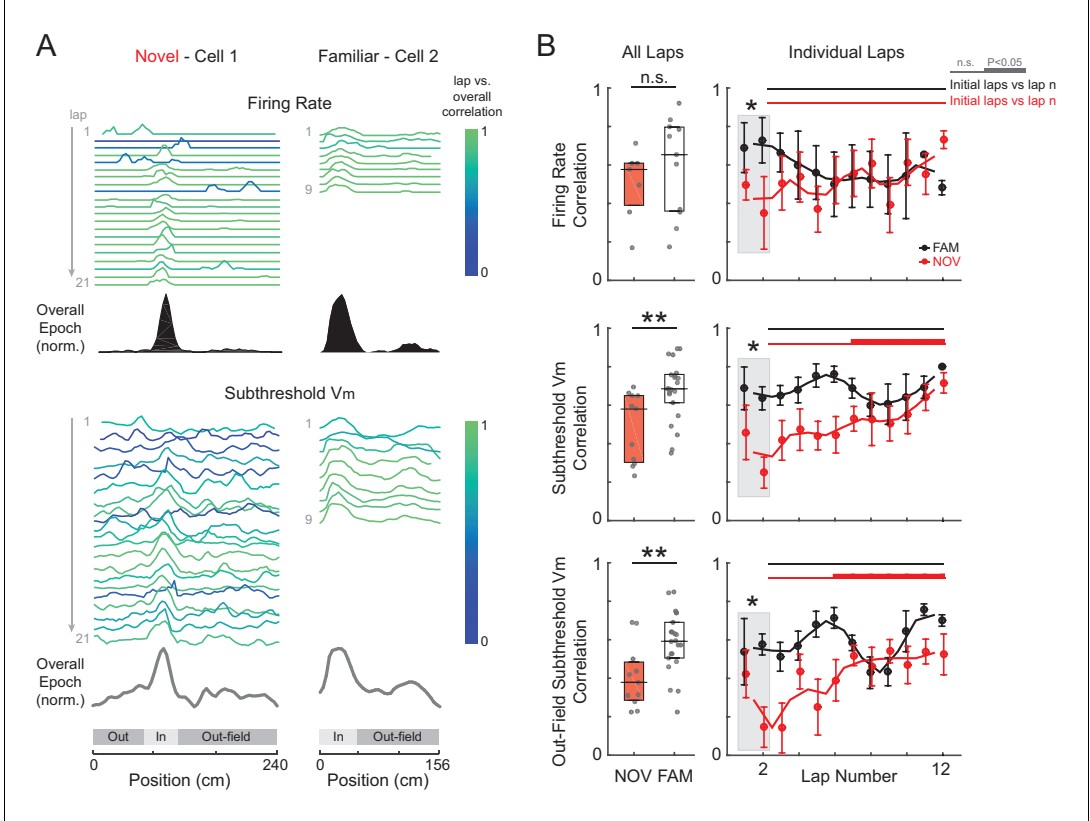

**Figure 5.** Spatial tuning of inputs throughout the entire environment stabilizes with experience. (**A**) Example of two cells recorded in a novel (left) and familiar (right) maze. AP rate (top) and subthreshold $V_m$ (bottom) activity (normalized to the peak value in each lap) for each lap, color-coded based on the spatial correlation score (Pearson's r) of the lap's AP rate or subthreshold $V_m$ profile with the overall epoch profile. (**B**) Left: Mean spatial correlation of the AP rate (top), subthreshold $V_m$ in all locations (middle), and out-field (i.e., excluding the $V_m$-defined field region) subthreshold $V_m$ (bottom), averaged across all individual laps shown in the corresponding panels on the right. This includes all cells recorded in a NOV or FAM maze (unpaired, minimum of 4 laps required). Right: Individual lap AP rate (top), subthreshold $V_m$ (middle), and out-field subthreshold $V_m$ (bottom) spatial correlation scores in NOV (red) and FAM (black) mazes (mean ± SE) versus lap number. Gray box highlights the comparison of NOV versus FAM within the initial laps (1-2). Black horizontal lines indicate the significance of unpaired comparisons (two lap smoothing) between the given FAM laps and the initial laps 1–2 in the FAM mazes; thick lines: $p<0.05$, thin lines: n.s. Similarly, red horizontal lines are for NOV mazes. Note that (**B**) excludes all first maze epochs experienced by the animal on that day (see text and *Figure 5—figure supplements 1* and *3*). *$p<0.05$; **$p<0.01$.

The following figure supplements are available for figure 5:

**Figure supplement 1.** Spatial tuning as a function of experience—additional examples and analyses, including the analysis of silent cells only.

**Figure supplement 2.** Stability of activity across laps in novel and familiar mazes, with examples of epochs that do and do not contain complex spikes.

**Figure supplement 3.** Excluding the initial maze epoch of the day does not alter the main findings of a novelty-induced enhancement of peak activity as well as other results.

**Figure supplement 4.** Position and shape of the subthreshold $V_m$ hill as a function of experience.

We addressed this question by analyzing the subthreshold $V_m$ outside of the peak subthreshold $V_m$ region ('out-field'). As with the overall spatial correlation scores, the out-field scores started low in the initial laps ($p=0.008$) of novel environments then increased in the later laps, and in familiar mazes were generally well-correlated across laps (*Figure 5B*, bottom). Furthermore, for the subset of epochs in which cells were silent, the spatial correlation of the subthreshold $V_m$ also increased during novel exploration and was high in familiar environments (*Figure 5—figure supplement 1C*). These results indicate that the increase in spatial tuning was not limited to the place field region,

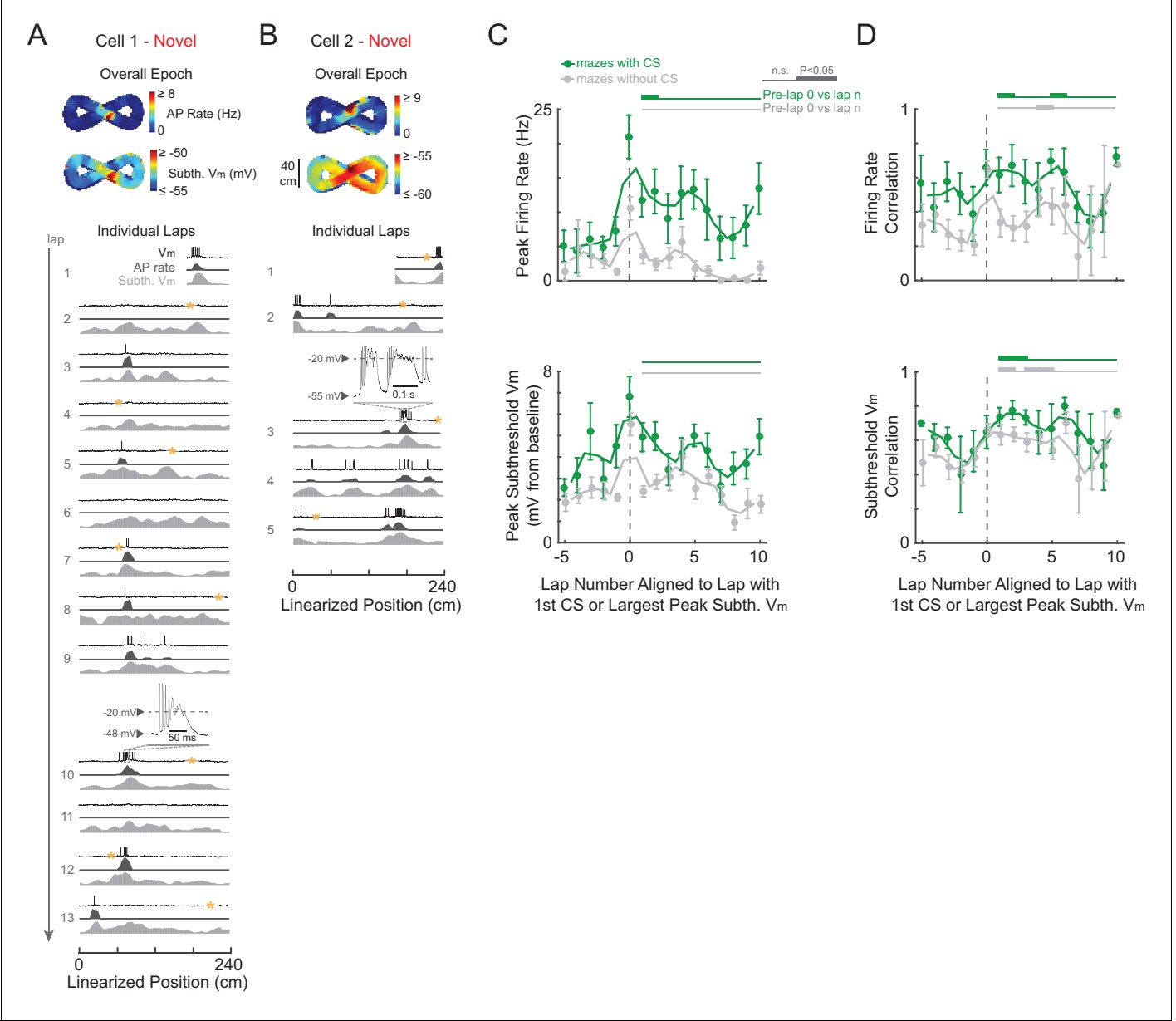

**Figure 6.** Complex spikes transiently boost spatial responses but are not required for place field formation in novel mazes. (A)-(B) Example novel maze recordings that contained ≥1 CS in the maze. Overall epoch 2-D AP rate map and subthreshold $V_m$ (above). Individual laps are shown as three traces: $V_m$ (top), AP rate (middle), and subthreshold $V_m$ (bottom), with the location and trace of the first CSs in each maze epoch highlighted. AP rate and subthreshold $V_m$ lap activity is normalized to the peak activity value in each lap. In no cases were CSs evoked by the current injections (asterisks) in (A)-(B). In (A), the place field had already formed before the first CS occurs. In (B), the CS occurs in the lap in which the place field first appears (same epoch as shown in *Figure 3C*). Note that there were no CSs in the novel maze place field recordings in *Figure 2B* or *Figure 3D*. (C) Peak AP rate (top) and subthreshold $V_m$ (bottom) for individual laps aligned (lap = 0) to the lap containing the first CS in the maze epoch for those epochs with a CS (green) or containing the largest peak subthreshold $V_m$ for those epochs without a CS (gray), then pooled across NOV and FAM maze epochs. Peak activity determined from the region (±3 spatial bins) surrounding the CS or peak subthreshold $V_m$ location. Mean ± SE. Green horizontal lines indicate the significance of paired comparisons (two lap smoothing) between the given laps and the mean of the pre-CS (pre-lap 0) laps; thick lines: $p < 0.05$, thin lines: n.s. Similarly, gray horizontal lines are for epochs without a CS. (D) Same as (C) for AP rate (top) and subthreshold $V_m$ (bottom) spatial correlation scores (Pearson's r).

The following figure supplements are available for figure 6:

**Figure supplement 1.** Properties of intracellular complex spikes in novel and familiar mazes.

*Figure 6 continued on next page*

*Figure 6 continued*

**Figure supplement 2.** Complex spikes in a familiar maze.

and that the spatial tuning of the rest of the inputs also stabilizes. This is consistent with stabilization of the entire network with experience, including excitatory inputs from hippocampal area CA3 and the entorhinal cortex (EC), each of which contain neurons that display spatially tuned firing (e.g., place fields in CA3 and grid cells in EC) that must itself be established in novel environments (*Leutgeb et al., 2004*; *Barry et al., 2012*).

While animals ran at similar speeds in novel and familiar environments, the spatial profile of running speed was somewhat less consistent during novel exploration (p=0.025; *Figure 5—figure supplement 1D*). However, the lower $V_m$ correlation scores in novel mazes were not attributable to differences in running behavior, as speed correlation scores did not predict $V_m$ correlation scores (p=0.28; *Figure 5—figure supplement 1E*; see Materials and methods).

We also measured the shape of the subthreshold $V_m$ hills. A backward shift of the hill with experience (i.e., a shift in location in the direction opposite to the animal's motion) (*Blum and Abbott, 1996*; *Mehta et al., 1997*) and a negative skew (i.e., a slower rise in the direction of the animal's motion) (*Mehta et al., 2000*; *2002*; *Ekstrom et al., 2001*; *Harvey et al., 2009*) are thought to result from spike timing-dependent plasticity (*Markram et al., 1997*; *Bi and Poo, 1998*). Signs of both were observed in the subthreshold $V_m$ hills underlying the AP place field (*Figure 5—figure supplement 4*), providing additional evidence of experience-dependent plasticity.

## Complex spikes are not necessary for place field formation in novel environments

Finally, we examined the role of CSs in place field formation and plasticity during novel environment exploration. CSs (also called plateau potentials in *Bittner et al., 2015*) are detected in somatic recordings as large amplitude, plateau-like $V_m$ depolarizations with accompanying spiking (*Figure 6A–B*, expanded traces), and are distinct (*Epsztein et al., 2011*) from extracellularly defined complex spikes (*Ranck, 1973*). CSs occurred in novel and familiar mazes and had similar properties in both (e.g., duration, number of APs, *Figure 6—figure supplement 1A–C*). CSs spontaneously occurred at low rates in both novel (14 CSs total in 10 of 190 total laps from nine maze epochs with ≥1 AP) and familiar (17 CSs in 11/228 laps from 26 epochs) mazes, were present in less than half of all cells (*Figure 6—figure supplement 1D*, left), and had occurrence rates that increased similarly with increasing subthreshold $V_m$ level (*Figure 6—figure supplement 1E*).

Even though CSs were rare events, previous work suggests that they play a significant role in the formation of place fields in novel mazes, as has been shown for familiar environments. Specifically, in previous work using a familiar treadmill-based virtual environment, CSs were always present in the first lap of spontaneously appearing place fields in cells that were silent in previous laps, and artificially evoked CSs were capable of immediately creating place fields in previously silent cells (*Bittner et al., 2015*). Therefore, we checked whether the initial appearance of place fields in novel mazes, which by definition are new place fields, was accompanied by CSs. Of the nine novel epoch place fields, four displayed stable activity in the place field region prior to any CSs (which then occurred later in the place field in 3/4 cases, e.g., *Figure 6A*), two had ≥1 CS in the initial lap of the emergent field (e.g., *Figures 3C and 6B*), and the remaining 3 were in epochs that did not have any CSs (e.g., *Figures 2B and 3D* did not contain CSs). Of these 9 place fields, 7 occurred in mazes that the animal had never previously been exposed to, and 2 occurred in mazes previously experienced 1 or 2 times (versus ≥19 exposures for familiar epochs, *Table 1*). Thus, CSs were not required for place field formation in novel environments. Moreover, CSs in familiar mazes did not necessarily stabilize place fields (*Figure 6—figure supplement 2*).

In addition to a specific role in forming new place fields in familiar environments (*Bittner et al., 2015*), CSs have been shown to trigger synaptic plasticity in general (*Takahashi and Magee, 2009*), so we examined the possibility that they contribute to the experience-dependent changes in activity that we observed. We checked whether CSs may contribute to the stabilization of activity by boosting subsequent activity and spatial correlation scores. We aligned each lap's activity to the lap containing the first CS in each epoch (whether novel or familiar). For comparison to epochs that did not

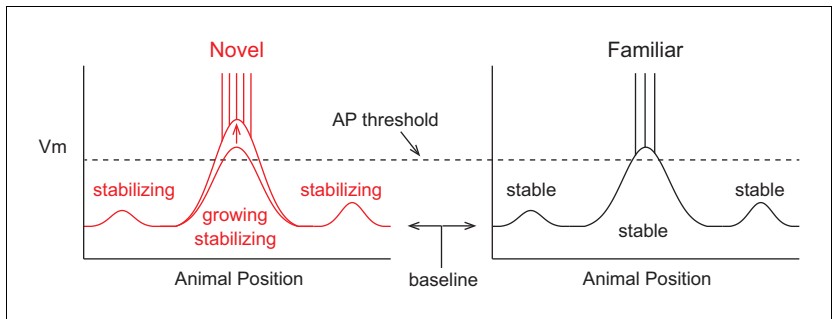

**Figure 7.** Summary of intracellular features underlying the formation of hippocampal CA1 place cell representations. Schematic based on findings from this study (compare to *Figure 1*). During novel exploration (left), the subthreshold $V_m$ depolarization underlying place fields grows in amplitude. This increase in amplitude is not sustained as the environment becomes more familiar (right). However, the spatial tuning of inputs, which is initially lower in all locations of novel mazes, stabilizes during novel exploration, both inside and outside of the place field. The higher AP rate in novel environments is due to the larger amplitude subthreshold depolarizations, as no differences were observed in the baseline $V_m$ or somatic excitability (e.g., AP threshold). CSs occur in a fraction of novel maze epochs and the appearance of place fields in novel environments does not require CSs. Thus, stable representations of familiar mazes are supported by the emergence of less strongly modulated, but more repeatable, spatially tuned subthreshold inputs.

contain CSs, we aligned laps in those epochs to the lap with the largest peak subthreshold $V_m$ (since CSs tend to occur at the most depolarized $V_m$ values, *Figure 6—figure supplement 1E*). We found a transient, but significant, increase in AP rate in the first lap after the first CS (*Figure 6C*, top), but not in subsequent peak subthreshold $V_m$ activity (*Figure 6C*, bottom). Regarding spatial tuning, we found a transient increase in AP rate and subthreshold $V_m$ spatial correlation scores after the first CS, though a similar result for the subthreshold $V_m$ spatial correlation was found following the largest peak $V_m$ for maze epochs without CSs (*Figure 6D*). Therefore, individual CSs appear to contribute an amount to plasticity and spatial learning that varies in magnitude – from small (i.e., transient increases in spiking and spatial correlation) to potentially large (i.e., occurring in the first lap of new place fields).

## Discussion

Using intracellular recording, we have investigated the mechanisms underlying how hippocampal spatial representations form and stabilize with experience. We began by testing a model of this process that was based on previous experimental and theoretical work (*Figure 1*). In summary (*Figure 7*), we found that the inputs driving place field spiking did indeed increase in amplitude during the exploration of a novel environment (*Figure 3*). Furthermore, this growth could occur before spiking occurred, suggesting a role for plasticity mechanisms that do not depend on postsynaptic spiking (*Golding et al., 2002*; *Dudman et al., 2007*). However, these increases in input amplitude were not sustained after additional experience in the environment, suggesting the need for alternate mechanisms of establishing stable place field activity in familiar environments. We then found an experience-dependent change that was maintained with familiarization – an increase in the similarity of the spatial tuning of inputs across the entire extent of the environment (*Figure 5*). This is a signature of the long-term stabilization of the hippocampal representation and provides a foundation for reliable spatial firing in familiar environments.

We found that complex spikes (CSs), which can induce synaptic plasticity (*Takahashi and Magee, 2009*) and are sufficient to trigger place field formation in familiar environments (*Bittner et al., 2015*), were not required for the appearance of individual place fields during novel exploration (e.g., *Figures 2B*, *3D* and *6A*). However, temporary increases in firing rate were seen in the laps subsequent to CS occurrence (*Figure 6C*). This suggests that CS-based plasticity could contribute to the long-term stabilization of place fields, perhaps in a similar fashion to other large, calcium-mediated events (*Sheffield and Dombeck, 2015*). This would be consistent with experiments in which

NMDAR-dependent plasticity is not necessary for the formation of place fields in novel environments (*McHugh et al., 1996*; *Kentros et al., 1998*), but rather for their persistence across days (*Kentros et al., 1998*). Note that CSs could be more potent triggers for place field formation in proximal cue-based environments (*Bittner et al., 2015*) versus the distal cue-based environments of our study.

While inputs and spiking were boosted during novel exploration, we found no differences in the baseline $V_m$ and somatic excitability between novel and familiar environments (*Figure 4A–D* and *Figure 4—figure supplement 1*). However, across different cells recorded in novel environments, firing rates were inversely correlated with AP threshold (*Figure 4E*), in agreement with previous work showing that cells with higher excitability are more likely to display place field activity during novel exploration (*Epsztein et al., 2011*; *Rich et al., 2014*).

Altogether, our findings support the following model of hippocampal spatial learning: (1) initially, small inhomogeneities in spatial inputs and excitability present during the first experience in an environment are amplified to produce robust place field activity without the need for plasticity; (2) plasticity during repeated traversals of the environment progressively refines and stabilizes this representation for long-term storage; (3) after learning, the inputs are no longer amplified, but the stability of these less strongly modulated inputs yields reliable spatially tuned firing. That is, the role of plasticity is to capture the initial representation and stabilize it before the network changes again, perhaps as a result of learning in other contexts in intervening periods. Plasticity would presumably stabilize CA1 place fields that happened to receive sufficiently strong, consistent, and/or matched (e.g., both EC and CA3) inputs (*Dudman et al., 2007*; *Sheffield and Dombeck, 2015*; *Bittner et al., 2015*; *Basu et al., 2016*). The initial expansion of activity (i.e., greater numbers of more active cells) could provide a diversity of synaptic and cellular patterns from which a sparser representation of space can be selected for long-term stabilization. The amplification of spiking activity could also serve to drive plasticity in downstream targets (*Lisman, 1997*; *Harris et al., 2001*; *Xu et al., 2012*) or send a general novelty signal (*Duncan et al., 2012*; *Larkin et al., 2014*) to the rest of the brain, which in turn may drive neuromodulatory feedback to the hippocampus and promote learning (*Lisman and Grace, 2005*). Afterwards, the comparatively low amplitude of inputs and firing rate in familiar environments may function to limit changes to the representation, further contributing to its stability. Additional changes, such as the creation of new fields via CSs (*Bittner et al., 2015*) at behaviorally important locations (*Hollup et al., 2001*; *Dupret et al., 2010*; *Monaco et al., 2014*), would presumably occur within a now stable spatial context. Variability of the low amplitude, but spatially reliable, modulation of inputs across the environment could explain why a given place cell is not always active in the same environment, but that when it fires, it does so in the same location (*Ziv et al., 2013*; *Rubin et al., 2015*).

What might underlie the amplification of inputs during novel spatial exploration? To begin with, there could be increased activity in input areas. However, the evidence to date indicates no difference in the average firing rates of excitatory neurons in CA3 (*Karlsson and Frank, 2008*) or EC (*Barry et al., 2012*) in novel compared to familiar environments. Whether there are differences in the activity of other excitatory inputs to CA1, such as the nucleus reuniens (*Herkenham, 1978*; *Vertes, 2015*), frontal cortex (*Rajasethupathy et al., 2015*), or the amygdala (*Sheth et al., 2008*) is currently unknown.

There are also multiple means by which the same excitatory input activity could result in larger amplitude depolarizations. Differences in the relative timing of inputs within and/or between, for example, CA3 and EC (*Dudman et al., 2007*; *Penley et al., 2013*; *Bittner et al., 2015*) could boost integration. The larger subthreshold theta oscillations we observed could reflect such altered timing, but could also simply be due to the increased peak $V_m$ values in novel mazes combined with the observed relationship between intracellular theta power and $V_m$ (*Kamondi et al., 1998*; *Lee et al., 2012*) (which could be caused by an increased driving force with respect to the inhibitory reversal potential). In addition, altered timing alone would not necessarily lead to the larger peak subthreshold responses we measured, as they involved the mean $V_m$ over many theta cycles.

Several types of changes in excitability could increase the size of the subthreshold response to a given input. First, a more depolarized somatic baseline $V_m$ can greatly enhance the subthreshold response of CA1 pyramidal neurons to spatial inputs (*Lee et al., 2012*, *2014*). However, as mentioned above, we found no evidence for differences in baseline $V_m$ (or input resistance) that could account for the larger peak subthreshold $V_m$ responses in novel mazes. While we did not detect

**Table 1.** Details of the individual cells and maze epochs. The 'P' and 'N' for epochs in mazes which the animal explored in both directions denote the two directions (i.e. moving in the direction of positive and negative changes in position value). The '#' indicates epochs that were excluded from analysis because their place fields appear to have been attributable to experimenter manipulations (e.g. inadvertent triggering of a CS during a current step used to probe series and input resistance and evoked spiking response), and thus their place field properties could not be assumed to be those of spontaneously occurring place fields. These epochs were however included in the analysis of **Figure 6C–D** since one can still assess the effect of any spontaneous CSs they displayed. The '##' marks an epoch in which the AP threshold could not be measured because all APs in the epoch contained a shoulder (see Materials and methods).

| Cell # | Epoch # | Maze | # Prev. Maze Sessions | Rec. dur. (s) | # Laps | Subth. vm baseline (mV) | Subth. vm peak (mV) | AP rate peak (Hz) | AP rate mean (Hz) | AP threshold (mV) | Input R. (MOhm) | AP rate Infield-Outfield ratio | Place field (0=No, 1=Yes) |
|---|---|---|---|---|---|---|---|---|---|---|---|---|---|
| 1 | 1 | 'O' | ≥39 | 106.5 | 7 | −61.3 | 1.4 | 0.00 | 0.00 | N/A | 147.5 | N/A | N/A |
| 1 | 2 | '8' | 0 | 282.7 | 17 | −60.3 | 2.4 | 0.00 | 0.00 | N/A | 197.0 | N/A | N/A |
| 2 | 1 | 'O' | ≥39 | 99.7 | 5 | −56.9 | 1.2 | 0.00 | 0.00 | N/A | 88.8 | N/A | N/A |
| 2 | 3P | 'l' | ≥19 | 209.0 | 3 | −54.0 | 2.8 | 0.00 | 0.00 | N/A | N/A | N/A | N/A |
| 2 | 3N | 'l' | ≥19 | | 2 | −53.7 | 2.1 | 0.00 | 0.00 | N/A | N/A | N/A | N/A |
| 3 | 1 | 'O' | ≥39 | 171.5 | 6 | −56.5 | 1.9 | 2.32 | 0.59 | −48.5 | 77.7 | 6.1 | 1 |
| 3 | 2 | '8' | 1 | 124.3 | 5 | −55.4 | 1.7 | 4.14 | 1.68 | −49.4 | 195.3 | 1.7 | 0 |
| 4 | 2 | '8' | 0 | 807.7 | 13 | −52.8 | 1.1 | 1.35 | 0.44 | −48.1 | 58.1 | 2.1 | 0 |
| 5 | 1 | 'O' | ≥39 | 115.5 | 5 | −62.6 | 1.3 | 0.00 | 0.00 | N/A | 120.3 | N/A | N/A |
| 6 | 1 | 'O' | ≥39 | 103.1 | 8 | −57.2 | 2.0 | 0.00 | 0.00 | N/A | 83.6 | N/A | N/A |
| 6 | 2P | 'L' | 0 | 587.7 | 12 | −53.3 | 2.3 | 2.67 | 0.78 | −40.4 | 23.9 | 4.5 | 1 |
| 6 | 2N | 'L' | 0 | | 12 | −53.6 | 1.7 | 0.89 | 0.38 | −40.4 | 23.9 | 7.1 | 1 |
| 7 | 1 | 'O' | ≥39 | 96.6 | 8 | −66.1 | 1.4 | 0.00 | 0.00 | N/A | 83.4 | N/A | N/A |
| 7 | 2 | '8' | 1 | 147.7 | 9 | −66.4 | 1.9 | 0.00 | 0.00 | N/A | 86.8 | N/A | N/A |
| 7 | 3P | 'l' | ≥19 | 124.4 | 5 | −66.6 | 3.0 | 0.00 | 0.00 | N/A | 69.0 | N/A | N/A |
| 7 | 3N | 'l' | ≥19 | | 5 | −65.6 | 2.0 | 0.00 | 0.00 | N/A | 69.0 | N/A | N/A |
| 8 | 1 | 'O' | ≥39 | 174.9 | 6 | −54.7 | 1.6 | 0.80 | 0.07 | N/A | N/A | 32.6 | 0 |
| 9 | 2 | '8' | 0 | 121.0 | 6 | −59.4 | 1.1 | 0.00 | 0.00 | N/A | 68.9 | N/A | N/A |
| 10 | 1 | 'O' | ≥39 | 143.9 | 5 | −57.0 | 2.0 | 6.58 | 1.36 | N/A | 128.0 | 3.9 | 1 |
| 11 | 1 | 'O' | ≥39 | 57.5 | 6 | −57.8 | 1.1 | 0.00 | 0.00 | N/A | 117.3 | N/A | N/A |
| 11 | 2# | '8' | 0 | 1058.3 | 9 | −55.7 | 7.9 | 12.05 | 0.89 | −42.2 | 112.7 | 35.5 | 1 |
| 12 | 1 | 'O' | ≥39 | 248.1 | 16 | −55.3 | 2.2 | 2.89 | 1.51 | −47.3 | 83.8 | 14.5 | 1 |
| 12 | 2P | 'l' | ≥19 | 67.5 | 4 | −55.9 | 3.9 | 9.52 | 2.57 | −46.0 | 73.6 | 3.7 | 1 |
| 12 | 2N | 'l' | ≥19 | | 3 | −55.6 | 3.4 | 14.08 | 4.14 | −46.0 | 73.6 | 3.3 | 1 |
| 13 | 1 | '8' | 0 | 1012.1 | 27 | −57.5 | 3.3 | 11.26 | 2.02 | −52.8 | 145.1 | 6.7 | 1 |
| 14 | 1 | 'O' | ≥39 | 224.8 | 13 | −54.9 | 1.7 | 2.10 | 0.80 | N/A | 69.0 | 1.5 | 0 |
| 14 | 2 | '8' | 1 | 367.1 | 21 | −55.6 | 3.6 | 8.46 | 0.75 | −48.0 | 55.4 | 25.7 | 1 |
| 14 | 3 | 'O' | ≥39 | 250.2 | 19 | −55.4 | 1.7 | 1.69 | 0.45 | −48.0 | 45.1 | 3.3 | 1 |
| 14 | 4 | '8' | 2 | 428.4 | 25 | −55.0 | 2.3 | 1.49 | 0.22 | −43.8 | 55.3 | 6.6 | 1 |
| 14 | 5P | 'l' | ≥19 | 203.5 | 9 | −53.9 | 2.2 | 0.22 | 0.02 | −45.3 | 100.9 | 20.0 | 0 |
| 14 | 5N | 'l' | ≥19 | | 9 | −53.9 | 1.7 | 2.31 | 0.22 | −45.3 | 100.9 | 34.3 | 1 |
| 15 | 1 | 'O' | ≥39 | 37.6 | 4 | −60.5 | 1.5 | 0.00 | 0.00 | N/A | 40.4 | N/A | N/A |
| 15 | 2 | '8' | 0 | 505.6 | 26 | −54.2 | 3.8 | 7.56 | 1.00 | −47.5 | 49.9 | 15.3 | 1 |
| 15 | 3 | 'O' | ≥39 | 199.5 | 18 | −47.8 | 1.9 | 3.02 | 1.08 | −43.7 | 57.4 | 3.0 | 1 |
| 16 | 1 | 'O' | ≥39 | 99.7 | 6 | −58.8 | 2.5 | 0.50 | 0.04 | N/A | 59.8 | 29.4 | 0 |
| 16 | 2# | '8' | 1 | 302.6 | 20 | −57.0 | 3.2 | 8.86 | 2.48 | −50.4 | 56.8 | 3.2 | 1 |
| 16 | 3 | 'O' | ≥39 | 198.2 | 8 | −58.8 | 3.9 | 3.79 | 0.45 | −51.5 | 49.9 | 13.9 | 1 |

*Table 1 continued on next page*

*Table 1 continued*

| Cell # | Epoch # | Maze | # Prev. Maze Sessions | Rec. dur. (s) | # Laps | Subth. vm baseline (mV) | Subth. vm peak (mV) | AP rate peak (Hz) | AP rate mean (Hz) | AP threshold (mV) | Input R. (MOhm) | AP rate Infield-Outfield ratio | Place field (0=No, 1=Yes) |
|---|---|---|---|---|---|---|---|---|---|---|---|---|---|
| 16 | 4# | '8' | 2 | 271.3 | 13 | −54.3 | 4.6 | 14.51 | 4.06 | −50.7 | 39.9 | 5.0 | 1 |
| 16 | 5 | 'O' | ≥39 | 306.0 | 7 | −57.4 | 3.0 | 2.67 | 0.55 | −50.6 | N/A | 5.5 | 1 |
| 16 | 6# | '8' | 3 | 421.1 | 17 | −61.5 | 2.9 | 3.45 | 0.42 | N/A | N/A | 23.3 | 1 |
| 17 | 1P | 'l' | ≥19 | 604.2 | 9 | −52.5 | 6.6 | 7.99 | 1.53 | −47.0 | 42.6 | 7.5 | 1 |
| 17 | 1N | 'l' | ≥19 | | 10 | −52.3 | 3.6 | 21.46 | 4.64 | −47.0 | 42.6 | 11.4 | 1 |
| 18 | 1 | 'O' | ≥39 | 47.3 | 4 | −61.2 | 1.7 | 0.00 | 0.00 | N/A | N/A | N/A | N/A |
| 19 | 1 | 'O' | ≥39 | 191.9 | 4 | −56.7 | 2.6 | 0.48 | 0.03 | N/A | N/A | Inf | 0 |
| 19 | 2 | '8' | 0 | 221.2 | 3 | −64.3 | 1.6 | 0.00 | 0.00 | N/A | N/A | N/A | N/A |
| 20 | 1 | 'O' | ≥39 | 154.7 | 9 | −61.6 | 1.2 | 0.00 | 0.00 | N/A | 140.6 | N/A | N/A |
| 20 | 2 | '8' | ≥19 | 65.2 | 3 | −63.8 | 3.0 | 0.00 | 0.00 | N/A | 196.1 | N/A | N/A |
| 21 | 1 | 'O' | ≥39 | 169.6 | 10 | −54.9 | 0.8 | 0.54 | 0.04 | N/A | 169.4 | 79.0 | 0 |
| 21 | 2 | '8' | ≥19 | 60.1 | 3 | −53.1 | 1.0 | 0.00 | 0.00 | N/A | N/A | N/A | N/A |
| 22 | 1 | 'O' | ≥39 | 149.2 | 9 | −57.2 | 1.6 | 0.23 | 0.01 | −44.6 | 76.8 | Inf | 0 |
| 22 | 2 | '8' | ≥19 | 137.9 | 5 | −55.6 | 1.7 | 0.00 | 0.00 | N/A | N/A | N/A | N/A |
| 23 | 1 | 'O' | ≥39 | 233.1 | 5 | −63.1 | 3.0 | 0.00 | 0.00 | N/A | N/A | N/A | N/A |
| 23 | 2 | '8' | ≥19 | 206.1 | 4 | −47.4 | 2.3 | 7.49 | 2.47 | −44.3 | 183.5 | 3.1 | 1 |
| 24 | 1 | 'O' | ≥39 | 295.6 | 8 | −48.2 | 1.6 | 3.96 | 1.41 | −46.4 | 159.3 | 2.7 | 0 |
| 24 | 2 | '8' | ≥19 | 384.7 | 5 | −49.6 | 5.8 | 14.50 | 2.73 | N/A | N/A | 5.4 | 1 |
| 25 | 1 | 'O' | ≥39 | 175.8 | 9 | −58.1 | 2.4 | 1.17 | 0.07 | −43.2 | 85.6 | Inf | 0 |
| 25 | 3P | 'L' | 0 | 1214.3 | 10 | −56.4 | 1.2 | 0.09 | 0.01 | −46.0 | 80.8 | Inf | 0 |
| 25 | 3N | 'L' | 0 | | 11 | −56.5 | 3.4 | 1.90 | 0.46 | −46.0 | 80.8 | 31.5 | 0 |
| 26 | 1 | 'O' | ≥39 | 53.2 | 3 | −60.6 | 2.5 | 0.00 | 0.00 | N/A | N/A | N/A | N/A |
| 27 | 1 | 'O' | ≥39 | 77.9 | 4 | −52.0 | 8.4 | 12.61 | 3.61 | −44.9 | 77.4 | 3.6 | 1 |
| 27 | 2 | '8' | ≥19 | 399.4 | 4 | −53.5 | 8.2 | 14.84 | 3.06 | −44.9 | N/A | 4.0 | 1 |
| 27 | 3 | 'O' | ≥39 | 105.2 | 8 | −56.4 | 3.8 | 4.26 | 0.81 | −46.6 | 67.1 | 4.3 | 1 |
| 27 | 4 | '8' | ≥19 | 112.8 | 6 | −57.8 | 3.9 | 0.57 | 0.03 | N/A | N/A | Inf | 0 |
| 28 | 2 | '8' | ≥19 | 217.2 | 4 | −55.4 | 2.5 | 0.76 | 0.06 | −41.4 | 145.7 | 23.3 | 1 |
| 29 | 1 | 'O' | ≥39 | 135.3 | 7 | −59.4 | 1.7 | 0.00 | 0.00 | N/A | 75.4 | N/A | N/A |
| 30 | 1 | 'O' | ≥39 | 141.7 | 6 | −60.9 | 1.2 | 0.00 | 0.00 | N/A | 43.4 | N/A | N/A |
| 31 | 2 | '8' | ≥19 | 82.4 | 3 | −56.0 | 1.5 | 0.00 | 0.00 | N/A | N/A | N/A | N/A |
| 32 | 2 | '8' | ≥19 | 159.9 | 5 | −55.2 | 2.5 | 4.66 | 0.82 | −48.3 | 124.4 | 4.6 | 1 |
| 32 | 3P | 'L' | 0 | 378.1 | 14 | −53.8 | 3.0 | 11.95 | 3.29 | N/A## | 145.1 | 3.1 | 1 |
| 32 | 3N | 'L' | 0 | | 14 | −53.7 | 2.7 | 5.43 | 1.55 | N/A## | 145.1 | 3.1 | 1 |

differences in somatic excitability, changes in inhibition (*Lovett-Barron et al., 2012*; *Royer et al., 2012*; *Milstein et al., 2015*) or intrinsic conductances (*Johnston et al., 2003*; *Johnston and Narayanan, 2008*; *Giessel and Sabatini, 2010*) in dendrites could alter the response to synaptic inputs. Neuromodulators such as acetylcholine (*Hasselmo, 2006*; *Teles-Grilo Ruivo and Mellor, 2013*) and dopamine (*Li et al., 2003*; *Kentros et al., 2004*; *McNamara et al., 2014*; *Takeuchi et al., 2016*) could underlie a novelty-specific enhancement of inputs. For instance, neuromodulators have been shown to affect the gain of receptive field responses (*Fu et al., 2014*). An analogous mechanism could be at work here, though with a difference; the gain change shown in recent work occurs when comparing active and quiescent behavior (*Chiappe et al., 2010*; *Maimon et al., 2010*; *Niell and*

*Stryker, 2010*; *Bennett et al., 2013*; *Polack et al., 2013*; *Fu et al., 2014*; *Schneider et al., 2014*). In our case, the difference in responsiveness would occur between two similar active behaviors – running at similar speeds in environments differentiated only by the degree of familiarity. On the other hand, unlike what could be expected from an overall change in gain, we did not observe a difference in the amplitude of the second highest subthreshold $V_m$ peaks in novel compared to familiar environments.

Instead of an amplification of inputs in novel environments, alternate mechanisms such as habituation or synaptic depression (*Xu et al., 1998*; *Kemp and Manahan-Vaughan, 2007*) could decrease response magnitudes across days (*Lim et al., 2015*). In such models (*Lim et al., 2015*), the larger response to novel versus familiar items occurs in a bottom-up fashion and thus could occur without a novelty-specific signal. In any case, the increasing lap-to-lap similarity of the subthreshold $V_m$ indicates that some form of plasticity underlies the formation of stable representations of familiar environments.

The correlation between spike rate and AP threshold in novel but not familiar environments (*Figure 4E*) is consistent with a model in which initial differences in intrinsic excitability bias which cells will be originally recruited into a given memory representation, with changes in excitability over time resulting in the allocation of different sets of cells to subsequent experiences (*Han et al., 2007*; *Zhou et al., 2009*; *Epsztein et al., 2011*; *Cai et al., 2016*; *Rashid et al., 2016*). In particular, the lack of correlation between firing rate and threshold in familiar environments suggests that, after the stabilization of a representation, inputs determine activity irrespective of excitability levels. Furthermore, the correlation between firing rate and baseline $V_m$ in familiar environments (*Figure 4F*) suggests that learning leads to the emergence of a cell assembly (*Hebb, 1949*) representing each maze, resulting in a spatially uniform subthreshold depolarizing bias for all place cells active in a given environment. In this case, the lower amplitude of the $V_m$ hills under place fields in familiar mazes may not represent weaker inputs. Rather, the inputs driving place fields may have remained strengthened, but the out-of-field inputs may also have strengthened with familiarization, thus reducing the amplitude of the hill relative to the baseline. Meanwhile, the higher firing rate of inhibitory neurons in CA1 in familiar environments (*Wilson and McNaughton, 1993*; *Frank et al., 2004*; *Nitz and McNaughton, 2004*) could reduce the $V_m$ everywhere, keeping the average cell's baseline $V_m$ similar to that in novel environments, thereby masking the presence of these strengthened synapses.

## Materials and methods

### Virtual Reality software and behavioral setup

Our custom virtual reality software was developed at the HHMI Janelia Research Campus as part of Janelia's open-source virtual reality software platform (Jovian). Our software suite, named 'MouseoVeR', is written in C++ and built from a number of open-source software components (Boost, Bullet, osgBullet, osgWorks, OpenSceneGraph, Collada, OpenGL, and Qt) that work together to generate a system allowing users to create configurable visual environments. The MouseoVeR code used in this study is available at https://github.com/JaneliaSciComp/CohenBolstadLee_eLife2017. Virtual maze environments were created using the open-source animation software Blender (www.blender.org) and rendered by MouseoVeR. Blender environments were rendered using three virtual camera objects located at a single point in space. One camera faced forwards and captured a field of view (FOV) of 75°. The two other cameras were identical but rotated ±75° to create a total azimuthal FOV of 225°. For display, the three images were rear-projected onto a 30 in diameter, ~40% translucent cylindrical screen to create a final image encompassing a total FOV of 20° below, 60° above, and ±112.5° lateral with respect to the location of head fixation (*Figure 2A*, top). The cylindrical screen surrounded the spherical treadmill – a large and hollow lightweight polystyrene sphere (16 in diameter, 65 g) resting on a bed of ten individually air-cushioned ping-pong balls in an acrylic frame (http://www.flintbox.com/public/project/26501/). To track the motion of the treadmill, two cameras separated by 90° were positioned at the equator and focused on 4 mm$^2$ regions under infrared light (modified from FlyFizz, https://openwiki.janelia.org/wiki/display/flyfizz/Home; *Seelig et al., 2010*). The cameras captured 30 × 30 pixel images of the treadmill surface at 4 kHz. Motion of the treadmill was computed from the accumulated differences in the images over time. A brief description of the MouseoVeR processing loop is as follows. In each iteration of the

rendering loop, MouseoVeR communicates with the treadmill's data server to retrieve the updated motion values since the last request. The values coming from the data server are defined as Euler angles in the server's coordinate space. For calibration, we created mappings between 180° rotations of the treadmill along each of its three rotational axes (roll, pitch, and yaw) in real-world space and the data server's coordinate space. For converting real into virtual behavior, we ignored any yaw rotations and extracted from pitch and roll values a rotational direction as well as a magnitude of rotation (which was converted into an arc length using the treadmill radius). Manually (i.e., animal-) controlled heading direction was derived from the treadmill's pitch-to-roll ratio by continuously sampling, smoothing (~500 ms), and converting it into a °/s turning rate. The motion vector was sent to the physics engine to compute collisions with virtual objects (e.g., maze walls), and the resulting virtual movement was rendered for display at a synchronized projector frame rate of 30 Hz.

## Behavioral training

Eight-to-twelve week-old male C57BL/6NCrl (Charles River Laboratories, Wilmington, MA) mice were used for all experiments. All procedures were performed in accordance with the Janelia Research Campus Institutional Animal Care and Use Committee guidelines on animal welfare. Prior to behavioral training, a stainless steel head plate – with a large central recording well to access the hippocampus bilaterally – was attached to the skull surface using light-cured adhesives (Optibond All-in-One, Kerr; Charisma, Heraeus Kulzer, South Bend, IN) and dental acrylic. Sites for future hippocampal CA1 craniotomies (bregma: −1.6 to −2.0 mm AP, 1.2 to 1.8 mm ML) were marked with a fine-tipped cautery pen (Medline Inc., Northfield, IL). Mice were allowed to recover for at least five days before behavioral training started. During recovery, the mice were given food and water ad libitum, and a saucer running wheel was placed in the home cage.

The day before starting behavioral training, mice were placed on water-restriction (1.0 ml/day). From this day onwards, the saucer wheel was removed from the home cage overnight. Body weight and overall health were checked each day to ensure the mice remained healthy over the course of the experiment (*Guo et al., 2014*). On day 1, mice were acclimated to the experimental setup and head fixation procedure over several short sessions (increasing from ~1 to 15 min). The head was centered and fixed atop the treadmill, with the eyes ~20 mm from the surface. This created a comfortable ~20° angle below horizontal between the skull surface and the neck-to-tail body line. A tube (2.0 mm O.D. thick-walled glass with silicone-coated tip) was positioned near the mouth for delivery of artificially sweetened water rewards (acesulfame potassium, 4 mM, Sigma-Aldrich, St. Louis, MO; ~2 µl liquid drop per reward). Initially, mice earned rewards by simply licking the reward tube, and then later, by producing forward motions on the treadmill of increasing distances. Mice were shown the initial virtual maze, but motion in the maze was disabled and their view was restricted to that of the start location.

On day 2, mice were exposed to the initial virtual maze for a total of ~1 hr over three training sessions (~10, 20, and 30 min durations). Four 1-D virtual mazes were used in the study (*Figure 2A*, middle). Two mazes were closed paths: an oval- (~175 cm length) and a figure-8-shaped track (~240 cm length). Two mazes were bidirectional: an approximately straight track (~160 cm in each direction) and an L-shaped track (~135 cm in each direction). Mice were trained to explore the virtual mazes by navigating in accordance with the geometry of the maze (e.g., heading to the right when the path veers to the right). Rewards were earned as the mice completed laps around the virtual environments. For closed path mazes, a single location was selected as the primary reward zone where two rewards were always delivered after completing a lap. In addition, ~1 reward was delivered per lap in a random location for additional motivation. For bidirectional mazes, the primary reward zone for each direction was located at the corresponding end of the track where either 1 or 2 rewards were dispensed. In addition, ~1 reward was delivered at a random location per direction. If a mouse consumed less than 1.0 ml of liquid across all training sessions in a day, a supplement of water was provided to ensure at least 1.0 ml total was received each day. In the early stages of training, head direction was partially automated by MouseoVeR to ensure that the mice could successfully navigate all regions of the mazes. As performance and skill on the spherical treadmill improved, we reduced the contribution of the automated heading component and correspondingly increased the manually controlled turning component. This increased the contribution of the subject's running behavior to the control of head direction.

On days 3–5, mice continued to explore the same virtual maze over 2–3 training sessions per day, for a total of ~1 hr of in-maze time per day. On day 6, mice were exposed to a second, unique virtual maze, in addition to the initial maze, which was now familiar. For days 6–10, mice were trained to explore and become familiar with the second maze and the transitions between the two mazes. On day 11, after a minimum of 5 days of training in each of the virtual mazes, mice were considered ready for experimental recordings. From day 11 onwards, each mouse explored familiar and novel mazes during recording on 3–4 days, interleaved with additional training sessions or rest days.

## Electrophysiology

On recording days, trained mice were anesthetized with isoflurane (~1.5%,~0.8 l/min flow rate) and placed in a modified stereotaxic frame that clamps the head plate. If this was the first recording day from that hemisphere, a craniotomy (~1 mm$^2$) was performed over dorsal CA1. Dura removal was performed with the assistance of collagenase (1 mg/ml in 1x PBS with 1.5 mM Ca, Collagenase type I, Sigma-Aldrich). Collagenase was applied for ~10–15 min, followed by application of bovine serum albumin solution (1 mg/ml in 1x PBS, Sigma Aldrich) for ~5 min, and finally rinsed with saline. A glass recording pipette (~2 MΩ) filled with saline was lowered into the brain and used to monitor the extracellular local field potential and unit activity in order to accurately map the depth of the dorsal CA1 pyramidal cell layer. Once the depth of the pyramidal layer was determined, the craniotomy was cleaned and rinsed with saline. To protect the brain during the recovery period, we created a sealed environment by filling a well surrounding the craniotomy with saline that was closed off with a cap lined with silicone (Kwik-Cast sealant, WPI, Sarasota, FL). All animals were mobile within a few minutes after being placed in their home cages. After a minimum of 1 hr of recovery time, mice were placed on the treadmill. Blind in vivo whole-cell recordings were obtained from the right or left dorsal CA1 pyramidal cell layer (*Lee et al., 2009*, *2014*) using recording pipettes (5–7 MΩ) filled with an intracellular solution containing (in mM) K-gluconate 135, HEPES 10, Na$_2$-phosphocreatine 10, KCl 4, MgATP 4, and Na$_3$GTP 0.3 (pH adjusted to 7.2 with KOH) as well as biocytin (0.2%). After a gigaseal was obtained, the seal was held for up to ~4 min with no positive pressure to let the network recover before breaking into the membrane. After achieving the whole-cell configuration, the VR display was turned on and the mice were free to explore the mazes for sweetened water rewards as on training days. However, on recording days, in addition to the previously explored familiar mazes, mice could be exposed to a novel, previously unencountered, maze. Current-clamp measurements of V$_m$ (amplifier low-pass filter set to 5 kHz) were sampled at 25 kHz. The animal's behavior in the virtual maze was sampled at 30 Hz (the projector frame rate). All data analyzed is from recording periods with no holding current or no more than −20 pA of negative holding current applied to the pipette. Recordings were not corrected for the liquid junction potential. All neurons included in this study had the electrophysiological characteristics of somatic CA1 pyramidal whole-cell recordings (*Lee et al., 2009*; *Epsztein et al., 2011*; *Lee et al., 2012*, *2014*). To minimize the number of animals used, we attempted to maximize the number of recordings per animal. Due to recording from multiple cells within a single animal across a window of ~5–10 days, we did not attempt to recover the histology of the recorded cells.

## Data analysis

All analysis was done using custom-written programs in Matlab. All novel versus familiar maze epoch comparisons as well as other comparisons (e.g., initial versus late lap amplitudes within epochs) were done using non-parametric paired (Wilcoxon signed-rank) and unpaired (Mann-Whitney U) tests assuming unequal variances, unless otherwise noted. Correspondingly, all numerical values in the text are reported as the median ± standard error of the median, unless otherwise noted, and box-plots show the 25–75%-iles and the median (horizontal black line). (Standard error of the median was computed as follows. From the sample population containing n samples, we drew n samples with replacement to create a sample group and then computed the median of the sample group. This was repeated 10,000 times to obtain 10,000 sample group medians. Finally, we computed the standard deviation of the sample group medians to obtain the standard error.) Pearson's linear correlation coefficient was used to assess correlations between features, with significance determined with respect to the hypothesis that there was no correlation. All P-values reported are 2-sided. A value of p<0.05 was defined as statistically significant. Only temporally adjacent novel and familiar

mazes were included in the analysis of NOV-FAM maze pairs. For example, if the order of presentation of mazes during the recording protocol was (1) FAM, (2) NOV, (3) FAM', and (4) FAM2, then 2 NOV-FAM maze pairs were considered for analysis: FAM-NOV and NOV-FAM'. Analysis was limited to data collected during periods when the animal's speed was $\geq$5 cm/s unless otherwise noted. The choice of dividing each epoch into early and late periods between laps 5 and 6 was based on previous extracellular studies showing that a substantial amount of the change in place field spiking activity during an epoch occurs within the first five laps (*Mehta et al., 2000*; *Ekstrom et al., 2001*).

Thirty-two cells were recorded from 15 mice navigating in virtual mazes. A maze was considered novel if it was the first day that the animal had encountered that maze (with the exception of 3 cases in which the animal had encountered the maze once on a previous day). For all familiar maze epochs, the animal had previously explored the maze a minimum of 19 times over the course of at least 5 days. Cells were active (where 'active' was defined as follows: peak AP firing rate as a function of location averaged over the maze epoch $\geq$0.5 Hz and spiking in $\geq$2 laps) in 47% of epochs in familiar mazes (21/45 epochs, 15/29 cells) and 69% of epochs in novel mazes (9/13 epochs, 8/12 cells). If active epochs were instead defined as those with an overall mean AP firing rate >0.1 Hz, the results were similar (NOV: 9/13, FAM: 18/45). The ~1.5 fold larger fraction of active epochs in novel environments agrees with the ~1.4 fold increased fraction of active dorsal CA1 pyramidal cells in rats exploring real-world novel versus familiar mazes (*Karlsson and Frank, 2008*).

## Subthreshold membrane potential and AP firing rate

The locations in the virtual mazes were linearized by collapsing them onto a long curve that went around the track, giving a 1-D representation of animal location. For display purposes, the beginning (position = 0 cm) and end (position =~135–240 cm) of the curve, which represent the same location, was chosen based on where the cellular activity was approximately the lowest across all time periods in the maze epoch (overall epoch average). Movement from zero to the end represents a full lap in the maze. The subthreshold $V_m$ trace was estimated from the raw $V_m$ trace as previously described (*Epsztein et al., 2011*; *Lee et al., 2012*). Briefly, all APs and any parts of the raw $V_m$ trace directly attributable to the somatic spikes themselves (e.g., after-depolarizations) were removed, as well as the entirety of the slow, large, putatively calcium-based depolarization that often follows a burst of APs (which thus includes removing the entirety of each CS). The remaining trace was then linearly interpolated across the resulting gaps, yielding the subthreshold $V_m$. The overall epoch mean AP firing rate and subthreshold $V_m$ as a function of the animal's virtual location were determined every 4 cm ('spatial bin') along the track, here using 12 cm-wide boxcar smoothing. The peak AP rate and peak subthreshold $V_m$ were defined as the maximum spatially binned values. The baseline $V_m$ (*Figure 4B*) was computed as the mean of the subthreshold $V_m$ values for the 10% of spatial bins with lowest subthreshold $V_m$. Similar operations were performed for individual laps to derive the AP rate, subthreshold $V_m$, and peak and baseline $V_m$ values for each lap. For each maze epoch, only directions in which the animal sampled each location on at least two different laps during the recording were considered for analysis. For bidirectional mazes, cellular activity in the direction that contained the largest overall epoch peak subthreshold $V_m$ response was selected for comparisons between NOV-FAM maze pairs. For analysis of place fields or activity across laps within an epoch, each direction was included independently. Some laps were eliminated from analysis because of holding current outside of the −20 to 0 pA range, or because of within-lap continuous baseline shifts (e.g., possibly due to residual dialysis even after waiting ~4 min after seal formation before breaking in, or from sudden brain movements relative to the pipette, etc.). The original absolute lap numbers were used for analysis when intervening laps were eliminated.

## Determination of the place field region and amplitude of the subthreshold $V_m$ hill

Place fields were determined for each epoch and direction as follows. From the epoch average firing rate map, the candidate place field was defined as the region containing the peak AP firing rate and contiguous bins with rate greater than or equal to the baseline rate plus 20% of the difference between the peak and baseline rate (where the baseline is the mean rate in the 10% of bins with lowest rate, which is generally 0 Hz). Then, candidate place fields with peak rate $\geq$0.5 Hz, at least 2 laps of spiking in-field, and an in-field/out-field average firing rate ratio $\geq$3 were classified as place

fields. Unless noted as referring to the place field, the field was determined from the overall mean subthreshold $V_m$ as a function of position in the epoch as follows. From this subthreshold function, the baseline $V_m$ (*Figure 4B*) was determined as was the spatial bin with peak subthreshold $V_m$. The set of contiguous position bins around this peak bin where the mean subthreshold $V_m$ was greater than or equal to the baseline $V_m$ plus 30% of the difference between the peak and baseline $V_m$ was defined as the inside of the field. The amplitude of the subthreshold hill was computed by subtracting the baseline $V_m$ from the peak value of the mean subthreshold $V_m$. Similar operations were performed for each lap to derive within-lap baseline (*Figure 4—figure supplement 1B*) and peak (*Figure 3—figure supplement 1A*,right and 1C) $V_m$ values.

### Determination of the spatial response amplitude of AP firing rate and subthreshold $V_m$ for experience-dependent analysis

The in-field peak spatial activity in each lap within a maze epoch (*Figure 3A and J*) was determined as follows. For each maze epoch the peak AP rate and subthreshold $V_m$ value in each lap were determined from the region surrounding (±3 spatial bins) the location of the overall epoch peak AP rate and subthreshold $V_m$, respectively. Individual lap peak values from this in-field region were then averaged across a specific set of laps (e.g., laps 1–2 for the 'initial laps' in *Figure 3A*), or averaged across all laps to obtain a mean value for the maze epoch (*Figure 3G–H*). Individual lap peak values were also determined irrespective of location ('lap peak'), then collapsed across laps to obtain a mean value for the maze epoch (*Figure 3—figure supplement 1B–C*). For peak subthreshold $V_m$ activity in each lap within a maze epoch with respect to the place field (*Figure 3B, E–F and I*), we took the peak subthreshold $V_m$ value in each lap inside the place field region determined from the overall epoch AP activity. Individual lap peak values from this place field region were then averaged across a specific set of laps (e.g., laps 1–2 for the 'initial laps' in *Figure 3B*), or averaged across all laps to obtain a mean value for the maze epoch (*Figure 3I*). In the case that one of the epochs in the maze pair (e.g. in *Figure 3I*) was active but did not have a place field or was silent, values for that epoch were computed with respect to the region surrounding (±3 spatial bins) the location of the overall epoch peak subthreshold $V_m$.

To assess the effect of CSs on peak spatial activity within a maze epoch (*Figure 6C*), for all cells recorded in a FAM or a NOV maze, and for maze epochs that contained a CS, we first determined the location (lap number and spatial bin) of the first CS in the epoch. Then for each lap, we determined the peak AP rate and subthreshold $V_m$ in the region surrounding (±3 spatial bins) the CS location. These lap peak activity values were then aligned to the lap (lap = 0) that contained the first CS. Similarly, for comparison with maze epochs that did not contain a CS, we determined the lap number and spatial bin containing the peak individual lap subthreshold $V_m$ value. Then for each lap, we found the peak activity values within the region surrounding this subthreshold $V_m$ value.

### Spatial correlation of the AP firing rate, subthreshold $V_m$, and speed profile for experience-dependent analysis

For *Figure 5*, *Figure 5—figure supplements 1A–C and* and *2*, for maze epochs in which there were at least 4 laps of activity, the stability of spatial tuning of the AP firing rate and, separately, subthreshold $V_m$ was determined for each lap as follows. A lap correlation score was determined by computing the Pearson's correlation coefficient between the activity across each location (i.e., each spatial bin in the maze) within the lap with the overall epoch activity. A mean maze correlation score was then determined by taking the mean of the individual lap scores. For bidirectional mazes, the direction with the largest amplitude overall epoch peak subthreshold $V_m$ response was selected for comparison.

To assess the effect of CSs on spatial tuning within a maze epoch (*Figure 6D*), for all cells recorded in a FAM or a NOV maze, and for maze epochs that contained a CS, we first determined the lap number of the first CS in the epoch. For each lap, the spatial correlation scores were computed as described above and aligned to the lap (lap = 0) that contained the first CS. Similarly, for comparison with maze epochs that did not contain a CS, we determined the lap number containing the peak individual lap subthreshold $V_m$ value and lap correlation scores were then aligned to that lap.

To assess the animal's speed profile for each lap (*Figure 5—figure supplement 1D*), the spatial correlation was taken between each lap's speed profile and the average speed profile for the overall epoch. For *Figure 5—figure supplement 1E*, for every pair of laps in a novel epoch, a spatial correlation between the subthreshold $V_m$ of the 2 laps and between the speed profile of the 2 laps was computed. Then the correlation of these values across all pairs of laps in an epoch was computed. The distribution (i.e., median) of the correlation values (one per epoch) across all epochs was then compared to 0. In a related analysis, the relationship between the instantaneous speed of the animal and the instantaneous subthreshold $V_m$ value was determined as follows. For each novel and familiar epoch, the correlation between instantaneous speed and corresponding subthreshold $Vm$ was determined for all contiguous timestamps associated with speed >5 cm/s (which is the same speed threshold as is used in all the analysis). The distribution of correlation values (one per epoch) across all epochs was then compared to 0.

## Spectral analysis and standard deviation of the subthreshold $V_m$

For *Figure 3—figure supplement 2*, each maze epoch (and each lap) was separated into periods when the animal was inside versus outside the field for that epoch. In the case of the theta (5–10 Hz) band, the subthreshold $V_m$ trace was first reduced to a sampling rate of 500 Hz via decimation. Short-time Fourier transform (STFT) methods were applied to compute the average power in the 5–10 Hz band. Note that ~0.25 s of data were excluded before and after each of the crossing times between the inside and outside of the field to eliminate contamination due to the temporal extent of the STFT window. In the case of the gamma (25–100 Hz) band, a 25–160 Hz band-pass and 60 Hz notch filter were applied to the subthreshold $V_m$ trace, the sampling rate was reduced to 2 kHz, and the STFT of the resulting trace was taken. Then, the periods 60 ms before to 60 ms after each AP and intracellular complex spike burst were removed, as well as 60 ms of data before and after each crossing time between the inside and outside of the field, and the power in the 25–100 Hz band was averaged over the remaining periods inside the field. For the standard deviation of the subthreshold $V_m$ trace, a 1–160 Hz band-pass and a 60 Hz notch filter were applied, and periods 17 ms before to 17 ms after each AP and complex spike burst as well as crossing times were removed. Then, the standard deviation over the remaining periods inside the field was computed.

## Determination of spontaneous AP threshold

The spontaneous AP threshold (*Figure 4A*) for a given maze epoch was determined as previously described (*Epsztein et al., 2011*). Briefly, for each AP, we set the threshold to be the $V_m$ value at which the dV/dt crossed 10 V/s (or 0.33 × the peak dV/dt of that AP, whichever was lower, in order to handle the slower APs that occurred later within bursts and CSs) on its way to the AP peak $V_m$. Note that the threshold for individual APs varies with the level of depolarization before the AP (*Figure 4—figure supplement 1A*), and therefore we determined the threshold from a subset of APs that did not have APs immediately preceding them (i.e., isolated spikes and first spikes of bursts), that occurred during less depolarized periods, that did not possess a dV/dt shoulder (*Epsztein et al., 2010*), and that were not a part of a CS.

## Determination of input resistance

During maze exploration, we injected two 100-ms-long hyperpolarizing current steps of −0.2 nA separated by 500 ms every 20 s through the recording pipette (*Figure 4C*, left). To determine the mean input resistance ($R_N$) within a maze epoch (*Figure 4C*), we first eliminated any $V_m$ responses to the current steps that were masked by large spontaneous fluctuations, averaged the remaining responses during the epoch, and then applied a previously described procedure (*Crochet and Petersen, 2006*) to the average response. The specific steps used were as previously described (*Epsztein et al., 2011*).

## Analysis of evoked number of APs

During maze exploration, we injected one 100-ms-long depolarizing current step of +0.1–0.2 nA every 20 s through the recording pipette (*Figure 4C*, left). For each maze epoch, a single stimulus intensity was chosen to evoke APs, and the mean number of APs evoked per stimulus was normalized to the median value across all maze epochs in which the same stimulus intensity was used. For

the comparison of evoked activity (*Figure 4D*), we only included data if the amplitude of the current step was the same for the FAM and NOV epochs of the maze pair.

## Shape of subthreshold $V_m$ hill under place field

For *Figure 5—figure supplement 4*, the overall epoch AP rate and subthreshold $V_m$ per lap as a function of position were determined every 2 cm along the track using 12 cm-wide boxcar smoothing for each epoch with a clear place field. From the AP rate function, the baseline was determined (mean of the AP rate values for the 10% of spatial bins with lowest AP rate, generally 0 Hz) as was the spatial bin with peak AP rate. The set of contiguous position bins around this peak bin where the mean AP rate was greater than or equal to the baseline plus 30% of the difference between the peak and baseline was defined as the inside of the AP rate field. The subthreshold $V_m$ from each lap was taken for positions starting one-fourth of the width of the AP rate field before the start of the (AP rate) field to one-fourth of the field width after the end of the field, then averaged across the desired set of laps (e.g., laps 1–2 for the 'initial' period). Each such curve was interpolated at 60 evenly spaced position values, normalized in amplitude (0 = minimum subthreshold $V_m$ value, 1 = maximum) and normalized with respect to the position along the field (0 to 1 for the inside of the field, −0.25 to 1.25 including the regions on each side of the field, and flipped if necessary so that the animal's running direction was from −0.25 to 1.25), then averaged across the different epoch's fields (mean ± SE).

## Complex spike occurrence

Classification of events as complex spikes (CSs) was done as previously described (*Epsztein et al., 2011*). Note that for the purposes of determining the location at which a CS occurred, we set the time of occurrence to be the time of the peak of the first AP in the CS. To assess the degree of association between novel place field formation and CSs, we considered all CSs whether or not they occurred when the animal was moving above or below the threshold speed (5 cm/s) and whether or not they occurred spontaneously or were triggered (i.e., evoked inadvertently by the 100-ms-long depolarizing current steps). In all other analyses involving CSs, we only included CSs that occurred spontaneously and when the animal was moving above the threshold speed.

## Acknowledgements

We thank Min Ji Kim for technical assistance; Joshua Dudman, Adam Hantman, Michael Tadross, Nelson Spruston, Jeffrey Magee, Aaron Milstein, Simon Peron, and members of the Lee Lab for helpful discussions and comments on the manuscript; Steve Bassin and Jason Osborne for mechanical engineering and machining; Vivek Jayaraman, Hannah Haberkern, Christopher Bruns, and Sean Murphy for contributions to the development of the virtual reality software; and Magnus Karlsson and Lakshmi Ramasamy for electrical engineering. This work was supported by the Howard Hughes Medical Institute.

## Additional information

### Funding

| Funder | Author |
| --- | --- |
| Howard Hughes Medical Institute | Jeremy D Cohen<br>Mark Bolstad<br>Albert K Lee |

The funders had no role in study design, data collection and interpretation, or the decision to submit the work for publication.

### Author contributions

JDC, Conceptualization, Software, Formal analysis, Investigation, Methodology, Writing—original draft, Writing—review and editing; MB, Software, Co-designed and wrote the virtual reality software; AKL, Conceptualization, Software, Formal analysis, Supervision, Methodology, Writing—original draft, Writing—review and editing

## Author ORCIDs

Jeremy D Cohen, http://orcid.org/0000-0003-4961-222X

Albert K Lee, http://orcid.org/0000-0003-4332-8332

## Ethics

Animal experimentation: All procedures were performed in accordance with the Janelia Research Campus Institutional Animal Care and Use Committee guidelines on animal welfare (protocol #14-116).

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
