## [Decision Letter]

Thank you for submitting your article "Experience-dependent shaping of hippocampal CA1 intracellular activity in novel and familiar environments" for consideration by *eLife*. Your article has been reviewed by three peer reviewers, and the evaluation has been overseen by a Reviewing Editor and Andrew King as the Senior Editor. The following individual involved in review of your submission has agreed to reveal his identity: Thomas J. McHugh (Reviewer #3).

The reviewers have discussed the reviews with one another and the Reviewing Editor has drafted this decision to help you prepare a revised submission.

Summary:

The present study uses intracellular recordings to investigate how environmental novelty is represented by hippocampal CA1 pyramidal neurons and what role plasticity plays in place cell formation and stabilization. The authors used a virtual reality (VR) setup to enable head-fixed mice running on a polystyrene sphere to switch between novel (NOV) and familiar (FAM) environments. The experimental design is clever and nicely controls for many behavioral variables. Overall, the study examines a clearly important and interesting question, and it is novel in a sense that this is the first of its kind to use intracellular recordings to study the effect of environmental novelty/familiarity on hippocampal CA1 place field formation and stabilization. Although the reviewers agree that there is considerable potential value in these findings, several major concerns were raised that will need to be addressed in any revision.

Essential changes:

1) These in vivo intracellular recordings are obviously challenging and state-of-the-art, since they would – ideally – require recordings from the same cells over different conditions (e.g. FAM>NOV>FAM, as in Figure 2) to collect data with robust, paired design and to control for animal-to-animal variability. Unfortunately, only a subset of the recordings in the study meet this criterion (n=11), and the authors had to include other cells with less complete recordings and with variable condition exposures. In addition, most of the recordings seem to be quite short (on average ~4 min/condition) and often contain only few laps (the laps/epoch reported is 10 +/- 7, e.g. Figure 4 and related supplementary figure). Obviously, statistical interpretability of any intracellular electrophysiological properties and experience-dependent changes during such short trials are in question. For example, place fields are not being defined in terms of statistical significance so this issue is not being factored in or controlled for. Furthermore, when analyzing stabilization of spatial tuning, the authors had to remove some epochs from FAM conditions (Figure 4) to show significant effects. Together, the data pools together cells with quite variable condition exposures/durations/quality and therefore it is quite difficult to assess at the moment what statistical power the dataset really possesses and the strength of conclusions that could be drawn from it. These issues need to be addressed.

2) A concern was expressed about the authors' place field definition, which is based on subthreshold membrane potential (V_m_). It is unclear how the authors can argue that a cell without spatially-tuned spiking activity could count as a bona fide place cell. This is a critical issue that should be carefully revisited, by focusing on cells with AP-based place fields to provide more informative insights into experience- and plasticity-dependent mechanisms of place field formation and stabilization and to better relate the results presented here to the existing literature. Since there is only a handful of cells the authors could really focus on here (i.e. cells with recordings in both FAM and NOV and with bona fide place fields at least in one condition), it would be important to also provide a table with recording duration, lap number, condition exposure, and physiological properties (e.g. sub/suprathreshold place fields) for each cell, which would allow a better assessment of the dataset.

3) An implicit assumption of the experimental design is that the animal can reliably discriminate between different VR conditions used to define NOV and FAM conditions. The key issue of the paper – novelty – is unfortunately not supported by any behavioral measures. The mice did not change their running speed or behavior (pausing) between the novel and familiar tracks. The authors note that the mice did show a change in the spatial profile of running (Figure 4—figure supplement 1), however it is unclear what this measure captures. Typically, mice show a significant change in running speed and distance traveled in a novel environment. While the physiological data seem to argue convincingly for a network response to novelty, it would be best if there was some measure of behavior that the authors could use as verification the animals are truly treating the virtual space as new. This is important also because the interpretation of experience-dependent changes in physiological properties of CA1 pyramidal cells would be quite complicated if context representations are largely overlapping in these VRs conditions and if partial or partial/rate remapping dominates the networks dynamics. In this case, one can argue, for example, that for cells with recording in only one (e.g. NOV) condition, place fields may have already formed during preceding FAM exposure(s) and then may or may not undergo some remapping in NOV. In general, the authors have to be clearer in their analysis on how many times they observe de novo place field formation.

4) What is the role of plasticity in creating new place cell representations? As a measure of plasticity, the authors use the in-field subthreshold membrane potential (V_m_) 'hill', defined as the peak value of the subthreshold V_m_ minus the baseline V_m_. One main finding of the study is that this subthreshold membrane potential hill in NOV continues to increase with the number of completed laps, while in FAM it does not. This effect seems quite robust, with ~1.7-fold increase in the subthreshold V_m_ hill leading to a 3-fold increase in firing rate. Overall the authors interpret this result in support of a model in which plasticity amplifies already existing place fields with repeated exposure to a novel environment until presumably novelty is reduced: "The increase in amplitude is consistent with a strengthening of inputs to place fields via Hebbian, spike timing-dependent, or other (Golding et al., 2002) mechanisms of synaptic plasticity, and indicates rapid learning in CA1 in novel environments." The authors' point is that inputs to place fields are strengthened, but the data show that this increase is not restricted to inputs within the place field, and this increase occurs irrespective of the peak V_m_ location. That is the lap peak V_m_ change appears comparable or even larger than the in-field V_m_ change (Figure 2), arguing against a field-specific plasticity mechanism. This may require more detailed analysis, again, by focusing on place cells with bona fide place fields. As mentioned above, defining place fields based on subthreshold V_m_ profile, instead of spiking, is different from conventional spike-based definition of place cells. Moreover, from this subthreshold definition of place fields it follows that the same (non-Hebbian?) experience-dependent plasticity mechanism would operate on both silent and active cells. Isn't it equally possible that instead of synaptic plasticity, these initial and transient changes in V_m_ during the first few minutes of a NOV exposure result mostly from non-specific neuromodulatory influences?

5) The authors argue in the last paragraph of the subsection “Spatial tuning of subthreshold V_m_ stabilizes more rapidly inside than outside the place field” (Figure 4—figure supplement 4) that the shape of the V_m_ hill shifts in a direction opposite to the motion of the animal through the place field, similar to what was observed with extracellular recordings in Mehta et al. 1997. The observation in the Mehta paper was not confined to novel tracks, however, at least in CA1, and seemed to reset daily, while the effect here, while not quantified, seems to only occur during the first exploration. This inconsistency should be addressed.

6) Because of the relatively short duration of the recordings, and because the only measure of novelty is the number of laps (and time) elapsed, which is arguably quite arbitrary (early vs. late), it is quite difficult to assess what exactly such short exposure to an environment means and what plasticity mechanism could be operational during this short time. Rather than asking about some arbitrarily defined 'initial' vs. 'late', it would be more informative, for example, to ask how changes in cellular dynamics correspond to behavioral dynamics (e.g. whether the peak amplitude is significantly different for laps before and after the mouse has learned its way through the novel environment – when it begins to reliably lick at the reward location).

7) The authors make some quite strong statements based on the data in Figure 4, but it is unclear that the data support such strong conclusions. The data in FAM environment are quite noisy to start with and, in addition, the authors need to exclude the first FAM epochs experienced by the animal on a given day: "All epochs were included, with the following exception. In our experiments, for each recording, the animal was generally first exposed to a familiar maze followed by a novel or other familiar maze. In many cases (10/32 cells), this first familiar maze recording was also the first exposure to any maze for the animal on that day. We suspected that spatial correlations could be disrupted in these first familiar epochs due to nonspecific behavioral factors, thus partially masking differences from novel epochs." This data selection need to be justified, since it treats NOV and FAM conditions differently and may introduce some artificial bias. For example, if there is a condition-independent, time-dependent stabilization during the initial laps in both NOV and FAM, such treatment may mitigate this effect for FAM but not for NOV, and exaggerate the experience-dependent changes in NOV. Importantly, the statistics in Figure 4 are difficult to interpret: the V_m_ stabilization for NOV-FAM and inside-outside the place field should not be analyzed with independent comparisons, but with 2-way ANOVA of context (NOV-FAM) and place field location (inside-outside). Can the authors identify significant effects for context, location and/or interaction this way? Alternatively, the authors could perform bootstrapping by repeatedly selecting regions outside the place fields on the track with equal width to the 'place field'.

8) Similarly, the complex spike (CS) part requires some more scrutiny. The authors state: "…4/7 epochs contained {greater than or equal to}1 CS in the initial lap of the emergent field (Figure 6), while 3/7 displayed stable activity in the place field region prior to any CSs (Figure 6) or did not have any CSs (for example, the novel maze place cell in Figure 2 did not contain a CS). Thus, CSs were not required for place field formation in novel environments…". So, it appears that >50% of place field were indeed formed following a CS, and in other cases place fields were already present prior to a CS – as appears to be the case in Figure 2 and Figure 6. How often were place fields actually formed de novo without a CS, which might argue against the role of CS in place cells formation. Again, how strong is the evidence for de novo place field formation in NOV? In other words, could it be the case that for place fields already present in NOV, these fields were formed during preceding FAM exposures? A more detailed analysis of the subset of cells with recordings in both FAM-NOV(-FAM) could perhaps be informative here.

9) Relatedly, the statement that "Moreover, CSs in familiar mazes did not necessarily stabilize place fields" should be qualified, since there seem to be no statistics supporting this claim other than the single example cells in Figure 6.

10) There is a confusing argument about where the stabilization of V_m_ occurred: in CA1 itself, or upstream in CA3 and Entorhinal cortex? It is unclear how an observation of different rates of correlation increase can be interpreted unambiguously as localizing the key plasticity steps to CA1.

---

## [Author Response]

Essential changes:

1) These in vivo intracellular recordings are obviously challenging and state-of-the-art, since they would – ideally – require recordings from the same cells over different conditions (e.g. FAM>NOV>FAM, as in Figure 2) to collect data with robust, paired design and to control for animal-to-animal variability. Unfortunately, only a subset of the recordings in the study meet this criterion (n=11), and the authors had to include other cells with less complete recordings and with variable condition exposures. In addition, most of the recordings seem to be quite short (on average ~4 min/condition) and often contain only few laps (the laps/epoch reported is 10 +/- 7, e.g. Figure 4 and related supplementary figure). Obviously, statistical interpretability of any intracellular electrophysiological properties and experience-dependent changes during such short trials are in question. For example, place fields are not being defined in terms of statistical significance so this issue is not being factored in or controlled for. Furthermore, when analyzing stabilization of spatial tuning, the authors had to remove some epochs from FAM conditions (Figure 4) to show significant effects. Together, the data pools together cells with quite variable condition exposures/durations/quality and therefore it is quite difficult to assess at the moment what statistical power the dataset really possesses and the strength of conclusions that could be drawn from it. These issues need to be addressed.

2) A concern was expressed about the authors' place field definition, which is based on subthreshold membrane potential (V_m_). It is unclear how the authors can argue that a cell without spatially-tuned spiking activity could count as a bona fide place cell. This is a critical issue that should be carefully revisited, by focusing on cells with AP-based place fields to provide more informative insights into experience- and plasticity-dependent mechanisms of place field formation and stabilization and to better relate the results presented here to the existing literature. Since there is only a handful of cells the authors could really focus on here (i.e. cells with recordings in both FAM and NOV and with bona fide place fields at least in one condition), it would be important to also provide a table with recording duration, lap number, condition exposure, and physiological properties (e.g. sub/suprathreshold place fields) for each cell, which would allow a better assessment of the dataset.

We have combined our responses to points 1 and 2 below. The reviewers raise questions related to the criteria for including cells in the analysis, especially regarding cells with (spiking) place fields versus active cells (that may not show spatially tuned spiking) or silent cells.

As the reviewers note, in several of our analyses on the relative magnitude of subthreshold V_m_ peaks, we included both active and silent cells and defined the field based on peaks in the average subthreshold V_m_ as a function of location. We believe these results using the subthreshold V_m_-defined fields of all cells are of value because they are unbiased regarding possible underlying mechanisms, such as plasticity mechanisms that do not require postsynaptic spiking (e.g. Golding et al., 2002; cited in the original manuscript for this possibility), and especially in light of our finding that the subthreshold V_m_ generally shows spatial tuning everywhere (whether the cell is active or silent). However, we fully agree with the reviewers’ point that these analyses should also be done with respect to spatially tuned spiking, and thus we defined place fields based on spiking and also analyzed the subthreshold V_m_ with respect to these place fields. The results agree with the original findings: growth of the subthreshold V_m_ within NOV and not within FAM epochs, and larger values in NOV compared to FAM. In addition, we further investigated the within-epoch growth of the subthreshold V_m_ and the results lead to a modified interpretation of the changes during NOV exploration. The details are as follows.

We classified cells as place cells (i.e. cells having a place field) by the following criteria for each epoch and direction: from the epoch average firing rate map, the candidate place field was defined as the region (“in-field”) containing the peak AP firing rate and any contiguous bins with rate above 20% of the way from the baseline to peak rate (where the baseline is the mean rate in the 10% of bins with lowest rate, which is generally 0 Hz); then candidate fields with peak rate ≥0.5 Hz, at least 2 laps of spiking in-field, and an in-field/out-field average firing rate ratio ≥3 were classified as place fields. As requested, we analyzed the NOV versus FAM peak subthreshold V_m_ data using only NOV-FAM pairs in which the cell had a place field in at least one maze epoch of each maze pair (i.e. place-place, place-active, and place-silent pairs). As before, the peak V_m_ responses were larger in NOV versus FAM epochs (NOV: 2.5 ± 0.4, FAM: 1.6 ± 0.1, n=10, P=0.004). We have added this result to the text and figures (Figure 3), and the place field definition to the Methods.

Next, as the reviewers requested, we specifically assessed the relationship of spiking activity to the within-epoch growth of the subthreshold V_m_ hill inside the spiking place fields of NOV place cells. For this analysis we added a new figure (Figure 3), which includes examples (Figure 3) that illustrate the analysis. First, we compared the in-field peak subthreshold V_m_ for the laps before any in-field spiking activity (“Pre-Active” laps) versus the laps starting from the first lap of in-field spiking activity (“Post” laps). (The Pre-Active laps in the Figure 3 examples are illustrated with shades of blue, and the Post laps by green and shades of red.) As expected, the peak subthreshold V_m_ increased sharply (though non-significantly, since only a subset of place fields first appeared after lap 1) for these cases in which in-field spiking activity first appeared after lap 1. Then, as a key test, we compared the in-field peak subthreshold V_m_ in the first lap with in-field spiking (“First-Active” lap, which is shown in green in the Figure 3 examples) versus all subsequent laps (“Post’” laps, which is colored with shades of red in Figure 3). This included cases where the field was already present in the first lap of the epoch. We found no significant change (First-Active: 4.0 ± 0.9 mV, Post’: 4.1 ± 0.5, n=9, P=1.0; Figure 3, right). Together, these results indicate that much of the growth of the V_m_ hill within NOV epochs occurs at the transition from silent to spiking activity, with no consistent increase after the first active lap. That is, here we did not find evidence of postsynaptic spiking-based strengthening of inputs. Rather, the results are consistent with plasticity mechanisms that do not require postsynaptic spiking (e.g. Golding et al., 2002). This result is a clarification of what is occurring during field formation in NOV environments, and we have revised the interpretation of what these findings may mean for plasticity mechanisms in the Results and Discussion sections.

Furthermore, as the reviewers suggested, we now provide a table (Table 1) containing descriptive statistics for all cells included in the study.

Regarding the analysis of spatial tuning in Figure 4 and the removal of some epochs (those that were the first epoch of the day), we address this in the response to point 7.

3) An implicit assumption of the experimental design is that the animal can reliably discriminate between different VR conditions used to define NOV and FAM conditions. The key issue of the paper – novelty – is unfortunately not supported by any behavioral measures. The mice did not change their running speed or behavior (pausing) between the novel and familiar tracks. The authors note that the mice did show a change in the spatial profile of running (Figure 4—figure supplement 1), however it is unclear what this measure captures. Typically, mice show a significant change in running speed and distance traveled in a novel environment. While the physiological data seem to argue convincingly for a network response to novelty, it would be best if there was some measure of behavior that the authors could use as verification the animals are truly treating the virtual space as new. This is important also because the interpretation of experience-dependent changes in physiological properties of CA1 pyramidal cells would be quite complicated if context representations are largely overlapping in these VRs conditions and if partial or partial/rate remapping dominates the networks dynamics. In this case, one can argue, for example, that for cells with recording in only one (e.g. NOV) condition, place fields may have already formed during preceding FAM exposure(s) and then may or may not undergo some remapping in NOV. In general, the authors have to be clearer in their analysis on how many times they observe de novo place field formation.

The reviewers ask for evidence that animals distinguished between mazes that were novel versus familiar. We address this point in two ways: an analysis of predictive licking behavior, and an analysis of measures that potentially reflect (global) remapping.

We assessed the licking behavior of the animal as a measure that could be associated with experience in the mazes, and found clear evidence of learning in the virtual mazes. In all mazes (and each direction for the subset of bidirectional mazes) there was a single, fixed reward zone (RDZ) where rewards were always delivered to the subject. We found significantly more predictive licking behavior (licking immediately before reward delivery in the RDZ) in FAM versus NOV mazes (“Pre-RDZ”) (Figure 2). We also found an increase in predictive licking from the initial laps (1-2) to the late laps (6-end) in all epochs (Figure 2—figure supplement 2). These results indicate that the animal learned the location of the RDZ with experience and licked accordingly, and thus that the animals could actively discriminate between NOV and FAM environments. This has been added to both the text and the figures (as described).

Regarding the reviewers’ question of remapping, we looked for evidence that the spatial tuning of neural activity did or did not differ between the NOV and FAM VR environments. In freely moving animals, cells can display rate remapping by firing in corresponding locations of different mazes situated within the same room. But our VR mazes were geometrically distinct arenas and thus there were no clear geometrically corresponding locations. Therefore, to attempt a comparable analysis in VR, we used the most salient common reference point, the primary reward zone, to define corresponding locations. Previous VR studies have shown that a substantial fraction of CA1 cells fire at a fixed distance from reward zones (Ravassard et al. 2013). If this was true across our different VR mazes, it would suggest that the spatially tuned activity in NOV mazes was not truly novel activity. Therefore, we correlated the spatial distribution of spiking activity (and, separately, the subthreshold V_m_) aligned to the reward zone for each recorded pair of NOV and FAM mazes. We did not find any evidence (P>0.05 for all comparisons) of corresponding activity across mazes. That is, there was no evidence of rate remapping-like processes, and instead the results were suggestive of global remapping in the novel environments. This analysis is now included in a new figure (Figure 2—figure supplement 1).

We address the reviewers’ question about de novo place fields in our response to point 8.

*4) What is the role of plasticity in creating new place cell representations? As a measure of plasticity, the authors use the in-field subthreshold membrane potential (*V_m_*) 'hill', defined as the peak value of the subthreshold V_m_ minus the baseline V_m_. One main finding of the study is that this subthreshold membrane potential hill in NOV continues to increase with the number of completed laps, while in FAM it does not. This effect seems quite robust, with ~1.7-fold increase in the subthreshold V_m_ hill leading to a 3-fold increase in firing rate. Overall the authors interpret this result in support of a model in which plasticity amplifies already existing place fields with repeated exposure to a novel environment until presumably novelty is reduced: "The increase in amplitude is consistent with a strengthening of inputs to place fields via Hebbian, spike timing-dependent, or other (Golding et al., 2002) mechanisms of synaptic plasticity, and indicates rapid learning in CA1 in novel environments." The authors' point is that inputs to place fields are strengthened, but the data show that this increase is not restricted to inputs within the place field, and this increase occurs irrespective of the peak V_m_ location. That is the lap peak* V_m_*change appears comparable or even larger than the in-field Vm change (Figure 2), arguing against a field-specific plasticity mechanism. This may require more detailed analysis, again, by focusing on place cells with bona fide place fields. As mentioned above, defining place fields based on subthreshold Vm profile, instead of spiking, is different from conventional spike-based definition of place cells. Moreover, from this subthreshold definition of place fields it follows that the same (non-Hebbian?) experience-dependent plasticity mechanism would operate on both silent and active cells. Isn't it equally possible that instead of synaptic plasticity, these initial and transient changes in* V_m_*during the first few minutes of a NOV exposure result mostly from non-specific neuromodulatory influences?*

We agree with the reviewers that it’s critical to analyze what happens to the subthreshold V_m_ within the spiking place field of place cells. This has been addressed in our response to points 1 and 2 above.

Regarding the increase in the “lap peak” V_m_ – which averages the peak V_m_ per lap irrespective of location – within NOV epochs, we included this analysis as a control for variability of the subthreshold V_m_ field since it was not known in advance how much the peak might jitter or systematically shift (e.g. due to learning) in location. We found that the direction and significance of the results were the same whether using the lap peak values or the field-based values. The magnitude of the values was larger using the lap peak version because it first takes the peak V_m_ per lap then averages, while the field-based peak V_m_ first averages (and thus smooths) the V_m_ over laps then takes the peak. We have added a comment about this (i.e. the calculation of lap peak values as a control for possible jitter) to the revised text and have also moved the corresponding figures to the figure supplements.

5) The authors argue in the last paragraph of the subsection “Spatial tuning of subthreshold V_m_ stabilizes more rapidly inside than outside the place field” (Figure 4—figure supplement 4) that the shape of the V_m_ hill shifts in a direction opposite to the motion of the animal through the place field, similar to what was observed with extracellular recordings in Mehta et al. 1997. The observation in the Mehta paper was not confined to novel tracks, however, at least in CA1, and seemed to reset daily, while the effect here, while not quantified, seems to only occur during the first exploration. This inconsistency should be addressed.

To clarify the issue, we have revised this figure (now Figure 5—figure supplement 4) to also show the location of the field’s center of mass (CoM), and have overlaid the early and late field shapes to better allow comparison. The CoM does indeed shift in the direction opposite to the motion of the animal within both NOV and FAM epochs, consistent with the previous findings of Mehta et al., 1997.

6) Because of the relatively short duration of the recordings, and because the only measure of novelty is the number of laps (and time) elapsed, which is arguably quite arbitrary (early vs. late), it is quite difficult to assess what exactly such short exposure to an environment means and what plasticity mechanism could be operational during this short time. Rather than asking about some arbitrarily defined 'initial' vs. 'late', it would be more informative, for example, to ask how changes in cellular dynamics correspond to behavioral dynamics (e.g. whether the peak amplitude is significantly different for laps before and after the mouse has learned its way through the novel environment – when it begins to reliably lick at the reward location).

In previous studies, it has been shown that a significant amount of the change in spiking with experience (e.g. Mehta et al., 2000; Ekstrom et al., 2001) occurs within the first 5 laps of an epoch, which is the reason we chose laps 1-5 as “early” experience and laps 6-end as “late” values. We have added this point to the text. Furthermore, behavioral measures, such as the newly added predictive licking rate described in the response to point 3 (newly added Figure 2—figure supplement 2), provide evidence of learning at the behavioral level within the first 5 laps. Together, this provides support for focusing on changes in neural activity between the initial laps and laps 6-end, and for considering plasticity mechanisms that act on these timescales to underlie rapid hippocampal learning.

7) The authors make some quite strong statements based on the data in Figure 4, but it is unclear that the data support such strong conclusions. The data in FAM environment are quite noisy to start with and, in addition, the authors need to exclude the first FAM epochs experienced by the animal on a given day: "All epochs were included, with the following exception. In our experiments, for each recording, the animal was generally first exposed to a familiar maze followed by a novel or other familiar maze. In many cases (10/32 cells), this first familiar maze recording was also the first exposure to any maze for the animal on that day. We suspected that spatial correlations could be disrupted in these first familiar epochs due to nonspecific behavioral factors, thus partially masking differences from novel epochs." This data selection need to be justified, since it treats NOV and FAM conditions differently and may introduce some artificial bias. For example, if there is a condition-independent, time-dependent stabilization during the initial laps in both NOV and FAM, such treatment may mitigate this effect for FAM but not for NOV, and exaggerate the experience-dependent changes in NOV. Importantly, the statistics in Figure 4 are difficult to interpret: the V_m_ stabilization for NOV-FAM and inside-outside the place field should not be analyzed with independent comparisons, but with 2-way ANOVA of context (NOV-FAM) and place field location (inside-outside). Can the authors identify significant effects for context, location and/or interaction this way? Alternatively, the authors could perform bootstrapping by repeatedly selecting regions outside the place fields on the track with equal width to the 'place field'.

The reviewers ask for justification for our exclusion of the first maze experienced by the animal on a given day in the spatial tuning analysis (previously Figure 4, now Figure 5).

First, the reviewers raise the possibility that there could be changes in neural activity in all first-mazes-of-the-day whether NOV or FAM, and that if first FAM mazes were excluded, these changes could be mistakenly attributed to the NOV condition specifically. In the original manuscript we wrote “With these first familiar epochs excluded … we observed that …” This was written incorrectly, as we excluded all first mazes of the day for the analysis in Figure 5 (previously Figure 4), not only the first FAM mazes. (It turned out to be the case that no first mazes of the day were NOV mazes, which still addresses the reviewers’ point that no such NOV mazes should be included if the corresponding FAM mazes are excluded.) We apologize for the mistaken sentence and resulting misunderstanding, and have corrected the text.

Second, an analysis of licking shows a difference in behavior between the first mazes of the day in FAM environments and all other FAM epochs. In particular, although predictive licking appeared intact in these first FAM mazes, the mice licked significantly less after the reward had been delivered (presented in the newly added Figure 2—figure supplement 2). This is consistent with our initial view that there may have been a reduced level of task engagement in the first mazes of the day that could affect spatial tuning non-specifically. We feel that this may be an important point to consider for future studies using virtual reality and have also added a comment about this in the text for researchers to consider in their future projects.

Note that when no epochs are excluded from the analysis (now Figure 5—figure supplement 1), most of the main findings of Figure 5 are also present. Specifically, the spatial tuning in the first laps was significantly lower in NOV compared to FAM epochs, the spatial tuning showed a significant experience-dependent increase in NOV epochs, and the spatial tuning started high and remained so during FAM epochs.

As the reviewers suggested, we also performed statistical analyses on spatial correlation scores using the two-way RMANOVA. First, for all maze epochs we performed a two-way RMANOVA for context (NOV vs. FAM) and location (all locations vs. Out-field – note that we do not perform correlations on data solely from the In-field region as the number of spatial bins is often too small for such a comparison; instead, we compare activity for all locations with Out-field locations that excludes the In-field bins). This revealed a significant effect of location (F_(1,38)_=20.00, P<0.001). Excluding first epochs of the day, we observed a significant effect of location (F_(1,31)_=18.98, P<0.001) and context (F_(1,31)_10.65, P=0.003). Next, for all epochs we performed a two-way RMANOVA for context (NOV vs. FAM) and experience (initial laps 1-2 vs. late laps 6-end). This revealed a significant effect of experience (F_(1,26)_=8.75, P=0.007). Excluding first epochs of the day, we observed a significant effect of experience (F_(1,21)_=21.91, P<0.001) and context (F_(1,21)_=2.54, P=0.011). These results justify a more detailed look at the effect of location and experience in the NOV and FAM epochs separately, such as that provided in Figure 5.

In conclusion, we believe the data clearly show that the spatial tuning begins lower in NOV than FAM mazes, and that the tuning grows with experience within NOV. Regarding our interpretation of these spatial tuning results, please refer to our response to point 10.

*8) Similarly, the complex spike (CS) part requires some more scrutiny. The authors state: "…4/7 epochs contained {greater than or equal to}1 CS in the initial lap of the emergent field (Figure 6), while 3/7 displayed stable activity in the place field region prior to any CSs (Figure 6) or did not have any CSs (for example, the novel maze place cell in Figure 2 did not contain a CS). Thus, CSs were not required for place field formation in novel environments…". So, it appears that >50% of place field were indeed formed following a CS, and in other cases place fields were already present prior to a CS – as appears to be the case in Figure 2 and Figure 6. How often were place fields actually formed de novo without a CS, which might argue against the role of CS in place cells formation. Again, how strong is the evidence for de novo place field formation in NOV? In other words, could it be the case that for place fields already present in NOV, these fields were formed during preceding FAM exposures? A more detailed analysis of the subset of cells with recordings in both FAM-NOV(-FAM) could perhaps be informative here.*

Regarding de novo place field formation, one line of evidence in favor of NOV place fields as being de novo fields is the absence of evidence for rate remapping (discussed in the response to point 3). In particular, this would argue against the idea that the field was originally formed in a preceding FAM environment and re-expressed in the NOV environment.

We also redid the lap-by-lap analysis on CSs and novel place field formation now using the place field definition added in response to points 1 and 2. First, we note that during this analysis, we found 2 cells (which together contained 4 novel epochs included in the original manuscript) in which experimenter manipulations appeared to have triggered place field formation. In particular, one cell, which had a high rate of spontaneous CSs, also had some CSs triggered by our short current steps, and these triggered CSs appear to have generated place fields in those locations in subsequent laps (as in Bittner et al., 2015). Including or excluding these epochs did not alter the significance of the results regarding growth of the subthreshold V_m_ hill in NOV or the higher peak subthreshold V_m_ in NOV vs. FAM (though it did change the precise values, as marked in the revised manuscript). But, we removed these epochs from our NOV group in the revised manuscript because their place fields do not clearly represent spontaneous novel place fields. Their exclusion did however result in the following changes (which have been incorporated into the revised text and figures): it reduced the difference in CS occurrence in NOV vs. FAM epochs to below significance, and it removed the lack of increase in out-field V_m_ spatial correlation within novel mazes. Of relevance for the current point 8, it also reduced the number of cases in which spontaneous (not triggered) CSs occurred in the first lap of a novel place field (because one of the now-excluded cells had a high propensity for spontaneous CSs, including CSs that spontaneously occurred in the first lap of place fields). The resulting numbers of novel epoch place fields and their relationship to CSs has been added to the revised text:

“Of the 9 novel epoch place fields, 4 displayed stable activity in the place field region prior to any CSs (which then occurred later in the place field in 3/4 cases, e.g. Figure 6), 2 had ≥1 CS in the initial lap of the emergent field (e.g. Figure 3 and Figure 6), and the remaining 3 were in epochs that did not have any CSs (e.g. Figure 2 and Figure 3 did not contain CSs). Of these 9 place fields, 7 occurred in mazes that the animal had never previously been exposed to, and 2 occurred in mazes previously experienced 1 or 2 times (versus ≥19 exposures for familiar epochs, Table 1). Thus, CSs were not required for place field formation in novel environments.”

That is, only 2 of 9 novel epoch place fields had a CS in the first lap in which spiking in the field appeared. Also, of the 3 place fields in novel epochs without a CS, there was a CS in a preceding FAM epoch in 1 case. Therefore, these results support the hypothesis that cells can form new place field activity in novel environments without CSs.

9) Relatedly, the statement that "Moreover, CSs in familiar mazes did not necessarily stabilize place fields" should be qualified, since there seem to be no statistics supporting this claim other than the single example cells in Figure 6.

In Figure 6, in which the spatial tuning scores of all epochs that contained CSs are aligned to the lap containing the CS, we did not observe a CS-specific effect on the subsequent tuning of activity in the maze. That is, the tuning was similarly, transiently increased following control laps which had a large V_m_ depolarization but no CSs. This suggests that individual CSs usually do not have a large effect on subsequent activity. Looking at individual FAM epochs, for the 6 FAM epochs with at least one CS, spiking activity was stable throughout the entire epoch in 5/6 cases. And of the 5 stable epochs, spiking activity was already present in the field location in the laps prior to the CS in 4/5 cases. In the 5th case, the CS occurred in the first active lap. For the one FAM epoch (shown in Figure 6 in the original manuscript, now in Figure 6—figure supplement 2) with unstable spiking activity, the 2 CSs in the 9th lap did not lead to the stabilization of spiking activity in the field. We noted this as a counterexample to stabilization of fields by CSs, especially since one of the CSs was of particularly long duration, and such long-duration, single CSs have been associated with spontaneous place field formation (Bittner et al., 2015). Since it is an isolated example, we have moved it to the figure supplement, where we have also included the above numbers for all FAM epochs with CSs in the legend. Note that the number of FAM cases for which it was possible to assess whether CSs could stabilize a field (or create a stable field) was 2 (since in the other 4 cases the CS occurred in fields that were already stable), and in 1/2 cases the CS did not do so.

10) There is a confusing argument about where the stabilization of V_m_ occurred: in CA1 itself, or upstream in CA3 and Entorhinal cortex? It is unclear how an observation of different rates of correlation increase can be interpreted unambiguously as localizing the key plasticity steps to CA1.

As mentioned briefly in the response to point 8, the exclusion of NOV epochs with likely non-spontaneous place field activity had the consequence of changing the original result (a lack of increase of in out-field V_m_ spatial correlation with experience in NOV epochs) to an out-field correlation increase that generally matches the NOV increase in spatial correlation for the entire environment (in-field + out-field). This removes the issue of differential rates of correlation increase and what this might mean for the locus of plasticity. As a result, we have modified the text accordingly.